# Cytoskeletal tension actively sustains the migratory T-cell synaptic contact

Sudha Kumari[1,2,*] (ID), Michael Mak[3,†], Yeh-Chuin Poh[1,3] (ID), Mira Tohme[4], Nicki Watson[5], Mariane Melo[1,2], Erin Janssen[4], Michael Dustin[6] (ID), Raif Geha[4] & Darrell J Irvine[1,2,7,8,**] (ID)

## Abstract

When migratory T cells encounter antigen-presenting cells (APCs), they arrest and form radially symmetric, stable intercellular junctions termed immunological synapses which facilitate exchange of crucial biochemical information and are critical for T-cell immunity. While the cellular processes underlying synapse formation have been well characterized, those that maintain the symmetry, and thereby the stability of the synapse, remain unknown. Here we identify an antigen-triggered mechanism that actively promotes T-cell synapse symmetry by generating cytoskeletal tension in the plane of the synapse through focal nucleation of actin via Wiskott–Aldrich syndrome protein (WASP), and contraction of the resultant actin filaments by myosin II. Following T-cell activation, WASP is degraded, leading to cytoskeletal unraveling and tension decay, which result in synapse breaking. Thus, our study identifies and characterizes a mechanical program within otherwise highly motile T cells that sustains the symmetry and stability of the T cell–APC synaptic contact.

**Keywords** actin cytoskeleton; immunological synapse; symmetry breaking; synapse mechanics; T-cell migration

**Subject Categories** Cell Adhesion, Polarity & Cytoskeleton; Immunology

**The EMBO Journal (2020) 39: e102783**

## Introduction

During an immune response, T cells form specialized junctions with cognate antigen-presenting cells (APCs) termed "immunological synapses" (Negulescu *et al*, 1996; Grakoui *et al*, 1999). Such synapses provide a stable platform where a number of T-cell surface receptors including T-cell receptors (TCRs), adhesion, and costimulatory receptors engage with counter-ligands on the APC surface (Negulescu *et al*, 1996; Miller *et al*, 2002; Dustin, 2008a). A symmetric, sustained synapse is a hallmark of T-cell activation, both *in vitro* and *in vivo* (Miller *et al*, 2002; Stoll *et al*, 2002; Mempel *et al*, 2004; Bajenoff *et al*, 2006; Germain *et al*, 2012). How T cells, which are otherwise highly motile (Bousso & Robey, 2003; Miller *et al*, 2003; Tadokoro *et al*, 2006), sustain prolonged synaptic contacts with APCs remains poorly characterized. This is a crucial gap in our understanding of T-cell biology since synapse lifetime is a critical determinant of T-cell activation and function (Hugues *et al*, 2004; Celli *et al*, 2007; Skokos *et al*, 2007; Zaretsky *et al*, 2017).

Integrins, the actin cytoskeleton, and calcium signaling are all known to play important roles in the initial formation of an immunological synapse (Dustin *et al*, 1997; Wei *et al*, 1999; Fooksman *et al*, 2010; Comrie & Burkhardt, 2016; Martin-Cofreces & Sanchez-Madrid, 2018), but their roles in the subsequent maintenance and eventual dissolution of the synaptic contact are unclear. For example, integrin activation and actin polymerization are essential processes for T-cell adhesion and synapse formation, but these processes could also terminate the synapse by engaging new adhesions and cellular protrusions, respectively, and thus driving T-cell migration away from the APC. Furthermore, T cells are inherently highly migratory and have pronounced F-actin-dependent lamellar undulations even in the synaptic phase (Sims *et al*, 2007; Roybal *et al*, 2013). These lamellar dynamics could promote polarization and lateral movement of the T cell (Mullins, 2010), thereby breaking symmetry of the synapse. F-actin along with the activated integrin lymphocyte function-associated antigen-1 (LFA-1) is organized into a ring-like structure in the early synapse, and this ring-like pattern is thought to promote synaptic junctional stability (Wulfing *et al*, 1998; Kaizuka *et al*, 2007; Babich *et al*, 2012). However, this F-actin/integrin ring undergoes continuous fluctuations and is therefore amenable to symmetry breaking (Sims *et al*, 2007; Mullins, 2010; Lomakin *et al*,

1   Koch Institute of Integrative Research, MIT, Cambridge, MA, USA
2   Ragon Institute of Harvard, MIT and MGH, Cambridge, MA, USA
3   Department of Mechanical Engineering, MIT, Cambridge, MA, USA
4   Division of Immunology, Boston Children's Hospital, Harvard Medical School, Boston, MA, USA
5   Whitehead Institute of Biomedical Research, Cambridge, MA, USA
6   Kennedy Institute of Rheumatology, University of Oxford, Oxford, UK
7   Department of Biological Engineering, MIT, Cambridge, MA, USA
8   Howard Hughes Medical Institute, Chevy Chase, MD, USA
    *Corresponding author. Tel: +1 617 253 0656; E-mail: kumars04@mit.edu
    **Corresponding author. Tel: +1 617 452 4174; E-mail: djirvine@mit.edu
    †Present address: Department of Biomedical Engineering, Yale University, New Haven, CT, USA

2015). In addition, genetic lesions in actin regulatory proteins such as the Wiskott–Aldrich syndrome protein (WASP) do not interfere with T cells adhering to and forming synaptic contacts with APCs, but result in highly unstable synapses (Cannon & Burkhardt, 2004; Sims *et al*, 2007; Thrasher & Burns, 2010; Calvez *et al*, 2011; Kumari *et al*, 2015). In summary, the role of the basic motility apparatus, especially the actin cytoskeleton, in sustaining already formed T cell–APC contacts is unclear and warrants further investigation.

Maintenance of the immune synapse is known to be influenced by T cell-intrinsic factors such as the strength of TCR signaling (Henrickson *et al*, 2008; Bohineust *et al*, 2018). However, the processes downstream of the cellular migration machinery that enable the transition from an arrested to a motile state are not clear (Hugues *et al*, 2004; Celli *et al*, 2007; Skokos *et al*, 2007; Shulman *et al*, 2014). One of the ways in which the TCR-associated actin dynamics could enforce synapse symmetry and stability is by regulating mechanical forces within the synapse. Polymerization of actin is known to generate forces (Ridley *et al*, 2003), and organization of the polymerized F-actin network tunes the magnitude of these forces (Fletcher & Mullins, 2010; Blanchoin *et al*, 2014). Notably, adhesion forces in T cell–APC conjugates are actin-cytoskeleton-dependent and continue to evolve after initial synapse formation (Hosseini *et al*, 2009; Lim *et al*, 2011; Bashour *et al*, 2014; Hu & Butte, 2016), raising the possibility that a specialized F-actin organization and associated forces may regulate maintenance of the synaptic contact.

We hypothesized that examining the actin cytoskeleton and associated forces in T cells during synaptic contact breaking would provide important clues to the mechanical design principles that T cells employ for maintaining the immune synapse. We studied T-cell cytoskeletal organization at different stages of activation using a combination of super-resolution imaging, genetic and pharmacological perturbations, micromechanical measurements, and computational simulations, employing both model APC-mimetic surfaces and physiological APCs. We found that specialized actin microstructures termed actin foci, which form within the interface following antigen encounter, generate and sustain intracellular tension within the T cell at the contact interface, and this tension actively sustains the synapse after its formation. This process relies on continuous nucleation of actin at TCR microclusters by Wiskott–Aldrich syndrome protein (WASP) and interaction of freshly polymerized actin filaments with myosin II. Myosin contractile activity generates and maintains high in-plane tension across the synaptic interface. This high-tension actomyosin network eventually breaks as activated T cells downregulate WASP, leading to immediate relaxation of cytoskeletal tension followed by synaptic unraveling and resumption of motility. Taken together, these results uncover a novel antigen-triggered biomechanical program that regulates synapse symmetry in primary T cells and highlight sub-synaptic actin architectures that sustain their interactions with APCs.

## Results

### Synapse maintenance is associated with actin re-organization, but not integrin activation or calcium signaling

The actin cytoskeleton, integrins, and intracellular calcium flux all play important roles in mediating initial T-cell motility arrest and formation of a symmetric synapse with antigen-presenting cells. We thus first examined how these factors impact the maintenance of pre-formed synapses. To model synapse formation and eventual disengagement, we seeded mouse primary naïve $CD4^+$ primary T cells (referred to as "T cells" henceforth) onto an anti-CD3/intercellular adhesion molecule-1 (ICAM-1)-coated coverslip (antigen-presenting surface, APS) and allowed synapses to form. We then assessed cellular polarization over time by recording the cells' morphologic aspect ratio, AR, because the contact interface elongation, reflected as increase in AR, is tightly linked to the motile state of the T cell (Hons *et al*, 2018; Houmadi *et al*, 2018; Mayya *et al*, 2018; Negulescu *et al*, 1996; Fig 1A, and Appendix Figs S1 and S2). That is, synapses are radially symmetric in their stable state and polarize to break symmetry thereby acquiring an elongated morphology prior to resumption of motility (Movies EV1 and EV2).

Within seconds, T cells seeded onto an APS spread to form a symmetric contact interface (Appendix Fig S1). Symmetric synapses persisted for ~ 10 min, followed by a transition period when the cells began polarizing and becoming motile (Negulescu *et al*, 1996; Hons *et al*, 2018; Houmadi *et al*, 2018; Mayya *et al*, 2018), reflected in an increase in AR (Fig 1B, Movies EV1 and EV2). Structured illumination microscopy (SIM) revealed the presence of peripheral actin-rich lamellipodia and punctate actin structures distributed across the synaptic interface (Fig 1C), termed "actin foci" (Kumari *et al*, 2015) in stable (5 min) synapse. As synapse began to polarize at later times (20 min), actin foci were reduced even though total F-actin levels in the interface remained constant (Fig 1C–E). Note that the foci quantification methodology we used [employing a Gaussian mask generated by a $1.6 \times 1.6$ μm$^2$ rolling window, as described previously (Kumari *et al*, 2015)] identified near-complete loss of foci as a ~ 65% reduction in foci intensity, due to residual signal contributed by the lamellipodial F-actin (Fig EV1). As T cells formed synapses with an APS, the key integrin adaptor protein talin accumulated rapidly at the periphery of the interface and talin levels remained constant at later times as cells began polarizing (Fig 1D). Activating integrins using $MnCl_2$ (Dransfield *et al*, 1992) after a stable synapse was established did not affect subsequent polarization/synapse breaking at 20 min, and inhibiting LFA-1/ICAM interactions using the small molecule inhibitor A286982 (Liu *et al*, 2000) blocked T-cell polarization, indicating that integrin activation is not necessary to sustain synapse symmetry and may in fact be required for symmetry breaking (Fig 1F and G). Notably, treatment of cells with A286982 was accompanied by a retention of actin foci at 20′, supporting a previous report that activated integrins may reduce F-actin accumulation in foci (Tabdanov *et al*, 2015). Under all of the treatment conditions, talin recruitment as well as the total F-actin was unchanged (Fig 1H left and middle graphs); however, higher actin foci levels were associated with reduced synapse polarization at later times (Fig 1H, right graph). Elevation of intracellular calcium is a key signal that triggers initial arrest and formation of adhesive contacts between T cells and APCs (Miller *et al*, 2004). However, similar to inhibition of integrins, blocking calcium signaling using BAPTA (Balagopalan *et al*, 2018) induced a retention of actin foci and synapse symmetry at 20′ (Appendix Fig S3). Thus paradoxically, integrins and calcium signaling, which are critical for initial T-cell arrest and formation of a symmetric synapse following TCR triggering, are unable to sustain pre-formed synapses. In contrast, actin foci were enriched in the interface of T cells forming

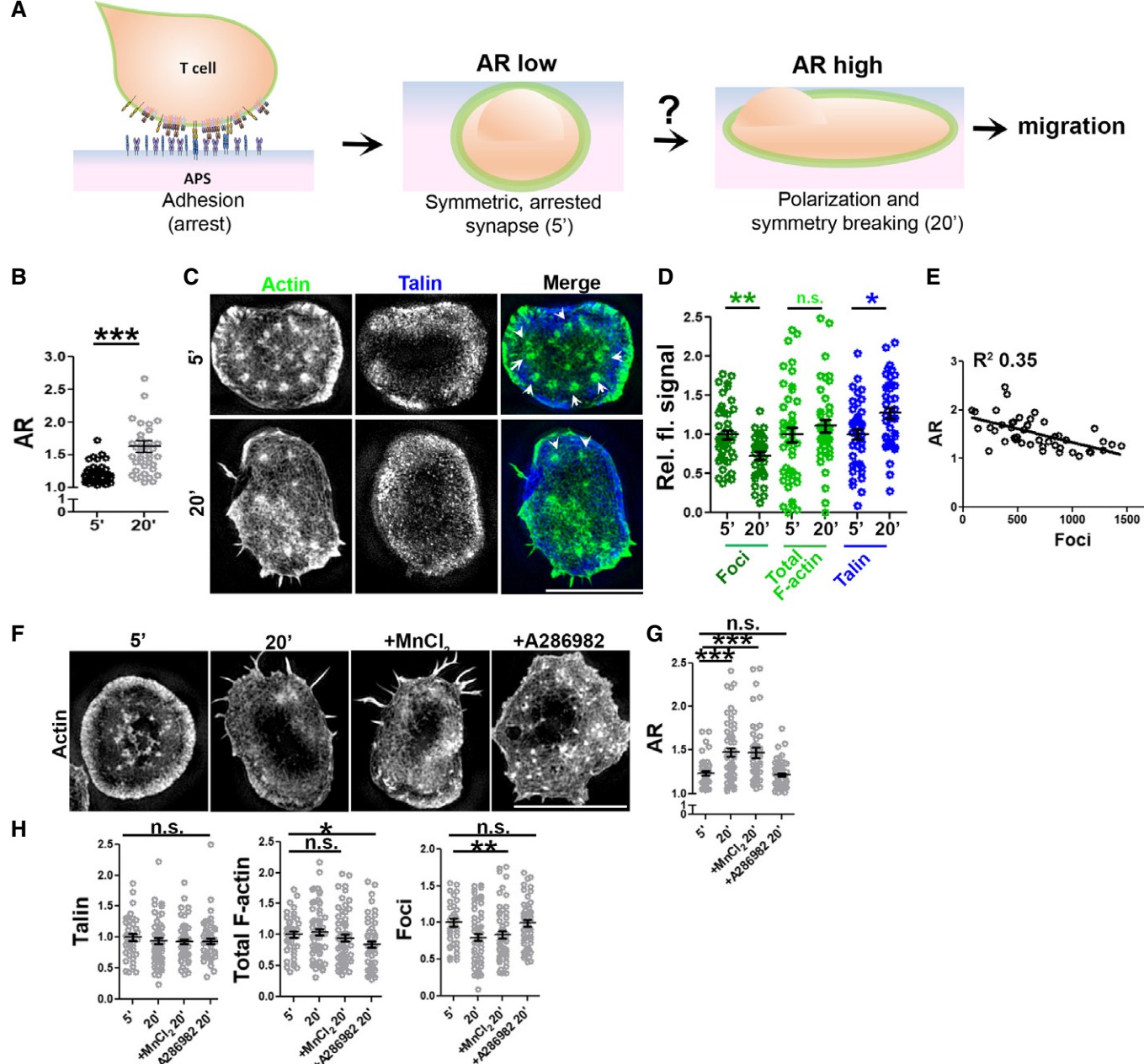

**Figure 1. Synapse polarization involves actin remodeling in mouse naïve CD4+ T cells ("T cells" henceforth) and proceeds despite of integrin augmentation.**

A    General schematic of the assay system used in this study for examining the process of synapse sustenance vs. polarization, post-initial T-cell adhesion, and spreading on the antigen-presenting surface (APS). Cell–APS interface of T cells was imaged for synapse establishment and maturation (5′) and eventual symmetry breaking and polarization (20′).

B–D   T-cell interface actin (phalloidin) re-organization during synapse symmetry breaking, imaged using SIM. Quantification of the shape elongation (AR; B, $n$ = 44 for 5′, 39 for 20′), or relative fluorescence signal of the indicated proteins in the SIM images (C, D) normalized to the mean of values at 5′; points represent values for individual cells. Arrowheads in (C) indicate foci. $P$ values are: ***$P$ < 0.0001 for AR, **$P$ = 0.003 for foci, **$P$ = 0.008 for talin, and $P$ = 0.36 for total F-actin using Mann–Whitney two-tailed test between populations of cells within the same experiment.

E    Relationship between AR and foci at individual cell level, each point in the scatter plot represents value obtained from a single cell.

F–H   Integrin augmentation does not rescue synapse symmetry breaking. Cells were allowed to adhere to the APS for 5′ or 10′ and were then treated with vehicle control, 0.5 mM $MnCl_2$ or 100 nM A286982 for subsequent 10′. Cells were fixed, stained with phalloidin and anti-talin antibody, imaged using SIM (F), and analyzed for AR (G) as well as talin, total F-actin, and actin foci at the synapse (H). In (E–H), $n$ for 5′ = 40, for 20′ = 62, for $MnCl_2$ = 59, for A286982 = 53. The values in the plots represent the intensity values normalized to the mean of 5′ in each set. $P$ values in the graph, n.s. > 0.05; ***$P$ < 0.001; *$P$ = 0.01; **$P$ < 0.009 using Mann–Whitney two-tailed test between populations of cells within the same experiment.

Data information: Central values in the graphs represent Mean, and error bars represent ± SEM. These experiments were repeated at least thrice with similar results. Scale bar, 5 μm.

symmetric synapses, and reduced or absent as T cells polarized and regained motility under all treatment conditions (Fig 1F–H).

## T-cell polarization is associated with WASP/actin foci downregulation

To directly visualize the dynamics of actin foci as T cells transitioned from the arrested to the motile state, we imaged LifeAct-GFP-expressing T cells using lattice light-sheet microscopy (LLSM). This technique exposes cells to lower phototoxicity than traditional microscopy (Chen *et al*, 2014), allowing long-term imaging of actin dynamics in naïve T cells which are highly sensitive to photodamage. LLSM imaging revealed that T cells engaged in a stable synapse continuously nucleated and dissipated actin foci across the contact interface, but as the cells polarized to begin migration, pre-existing foci were lost and new foci ceased to form (Fig 2A and B; Movie EV3). Thus, the loss of actin foci accompanies synapse symmetry breaking.

We posited that the loss of actin foci during synapse disengagement may result from downregulation of WASP since actin foci at the synapse are nucleated by WASP (Kumari *et al*, 2015) and TCR signaling-induced degradation of WASP has been previously reported (Macpherson *et al*, 2012; Reicher *et al*, 2012; Watanabe *et al*, 2013). Consistent with this hypothesis, we detected a sizeable reduction in total as well as active forms of WASP (phospho-Y293) at 20′ post-T-cell activation. WASP downregulation correlated with diminution of actin foci at the T cell–APS synapse as revealed by TIRF microscopy and Western blotting (Fig 2C–E). Similar correlations between WASP downregulation with reduction in actin foci and onset of cellular morphological polarization at 20 min were found when examining T cell–bone marrow-derived dendritic cell (BMDC) synapses using confocal microscopy (Appendix Fig S4). When the stable synapse was imaged at high resolution, highly dynamic lateral protrusions and retractions at the cell periphery (lamellar fluctuations; Roybal *et al*, 2013) were visible along with continuously nucleated foci (Fig 2F–H, Movie EV4). Comparison of foci and lamellar dynamics showed that foci had a mean half-life ~ 3 times longer than the lamellar actin fluctuations in WT cells (Fig 2H). In WASP$^{-/-}$ cells lacking foci, lamellar fluctuations dominated the overall F-actin dynamics at the synapse (Fig 2I, Movie EV4), as expected. Thus, even during the stable phase of the arrested synapse, actin polymerization drives constant lamellar fluctuations in the periphery that could drive symmetry breaking (Carvalho *et al*, 2013; Blanchoin *et al*, 2014), while actin foci are dynamically nucleated across the interface.

## Synaptic actin foci dynamics and associated traction forces support synapse symmetry

Continuous nucleation of actin filaments at the foci can, in principle, generate localized mechanical stresses (Cossart, 2000; Pollard & Borisy, 2003). To assess whether this was the case here, and whether the resulting forces may help sustain synapse symmetry, we quantified foci-dependent forces in the synaptic contact interface using traction force microscopy (TFM, Fig 3A; Engler *et al*, 2004; Yeung *et al*, 2005; Poh *et al*, 2012). Using this approach, we found that WT T cells exerted substantial forces on the substrate (Fig 3B). These forces were greatly diminished in WASP$^{-/-}$ cells that lacked foci (Fig 3B). Furthermore, the net contractile moment (pJ), a metric

describing force magnitude as well as distribution across the cell contact interface (Butler *et al*, 2002; Wang *et al*, 2002), showed that there were higher opposing forces separated by longer distances in WT synapses (pJ 0.35 ± 0.17) compared to WASP$^{-/-}$ synapses (pJ 0.06 ± 0.033; *P* value 0.01 between WT and WASP$^{-/-}$).

We further confirmed the link between foci and mechanical stresses at the synapse by determining their localization with the phosphorylated form of the mechanosensory protein CasL (pCasL) —a molecular marker of localized cytoskeletal tension. CasL interacts with the actin cytoskeleton and undergoes a conformational change in response to local cytoskeletal tension allowing stretching and phosphorylation of its N-terminal domain (Sawada *et al*, 2006; Janssen *et al*, 2016; Santos *et al*, 2016; Fig EV2). This conformational change can be detected by an anti-pY249 p130Cas antibody. At the synapse, pCasL significantly colocalized with actin foci and the phosphorylated form of Zap70 (pZap70), a TCR localized activation marker (Figs 3C, and EV2B and C). pCasL levels at synapse were reduced by the actin-cytoskeleton-disrupting agent latrunculin A (Fig EV2D), as expected from the cytoskeletal tension-dependent phosphorylation of CasL. WASP$^{-/-}$ T cells lacking actin foci, as well as polarized WT T cells, showed lower levels of pCasL at the interface at 20 min (Figs 3C and EV2E), confirming lower cytoskeletal stresses in these cells. Lower pCasL levels were not due to defective CasL recruitment at synapse in WASP$^{-/-}$ cells, since recruitment of CasL to the synapse was unaffected in these cells (Fig EV2F). Overexpression of a GFP-tagged form of WASP in T cells could rescue WASP levels, foci, pCasL, and AR in "late" synapses at 20 min (Appendix Fig S5A and B). Further, a reduction in pCasL during synapse polarization was not restricted to murine cells and was also observed in human primary CD4$^+$ T cells (Appendix Fig S5C).

When activated using the same activating substrates, synapses of WASP$^{-/-}$ cells polarized and gained motility faster than WT cells (Fig EV3B; Movie EV5), even though WASP$^{-/-}$ cells adhered better to the substrate and showed the same initial spreading kinetics and symmetrical morphology as WT T cells (Fig EV3A). The instability in WASP$^{-/-}$ synapses was not due to grossly perturbed antigen receptor signaling, since various key features of early TCR signaling were preserved in them (Fig EV4A). In fact, actin foci were not associated with signaling molecules Zap70 and PLCγ1 in late stages of synapses, indicating that once early signaling has been triggered (Kumari *et al*, 2015; Hashimoto-Tane *et al*, 2016), actin foci may be dispensable for TCR signaling (Fig EV4B and C). Unstable WASP$^{-/-}$ synapses were also not due to reduced integrin signaling, as WASP$^{-/-}$ cells did initially engage ICAM-1 at their synaptic interface and recruited normal amounts of talin (Appendix Fig S6A). Furthermore, treatment with MnCl$_2$ (Dransfield *et al*, 1992; Appendix Fig S6B) or supplementation of cytoplasmic Ca$^{2+}$ levels using acute treatment with thapsigargin (Gouy *et al*, 1990; Appendix Fig S7) did not prevent early polarization of WASP$^{-/-}$ cells, indicating that integrin activation and calcium signaling are insufficient to reconstitute synapse symmetry in these cells, as was the case with their inability to sustain synapse in WT T cells (Fig 1F–H, Appendix Fig S3).

To discount the possibility that lower mechanical stress at the synaptic interface and faster symmetry breaking in WASP$^{-/-}$ cells was an artifact of the minimal activation surfaces used here, or is a feature specific to murine T cells, we utilized naïve CD4$^+$ T cells derived from human WAS patient cells and activated them using

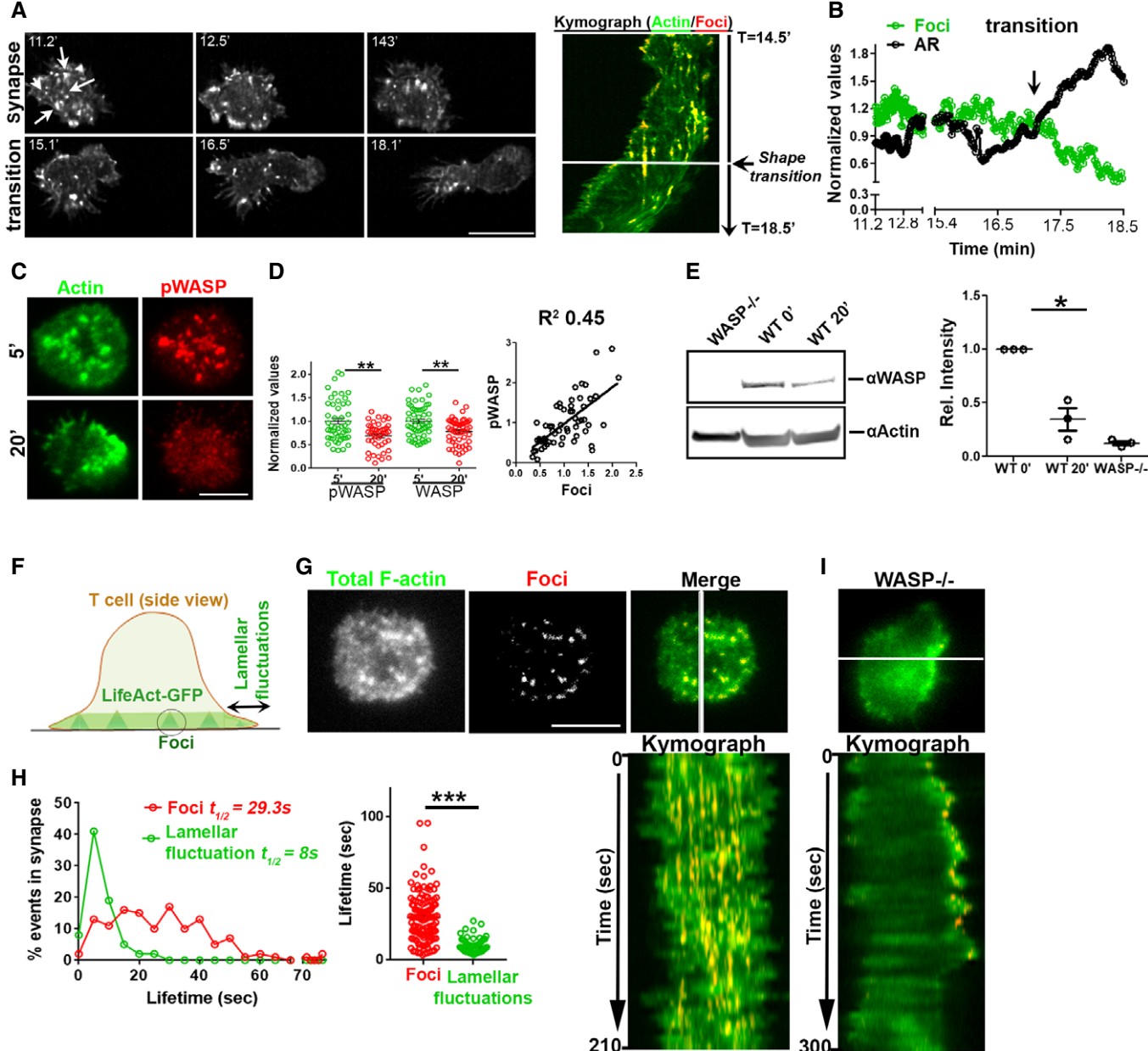

**Figure 2. Synapse symmetry involves temporal regulation of WASP and WASP-dependent actin kinetics.**

A    Time-lapse imaging of LifeAct-GFP-expressing T cell using lattice light-sheet microscopy (LLSM) during synapse polarization. Shown are snapshots at selected time points (left) and a kymograph of actin foci (denoted by arrows in the first image panel) over time (right) for a representative T cell.

B    Quantification of changes in actin foci and cellular aspect ratio in LLSM images during symmetry breaking. The values in the plot are the raw values normalized to their mean in each case.

C–E    T cells incubated with APS for the indicated time periods were processed for phalloidin or immune staining (C, D) or for Western blotting (E). Graph in (D) shows relative levels of total and active WASP (WASP-phosphorylated at Y293, pWASP), normalized to the mean levels at 5′; for pWASP *n* in 5′ = 49, in 20′ = 44; for WASP, *n* in 5′ = 56, 20′ = 56. *P* values, $P^{**}$ = 0.0027 for pWASP and $P^{**}$ = 0.0035 for WASP using Mann–Whitney two-tailed test. The scatterplot in (D) shows the relationship between active WASP and foci on a per cell basis. Numbers in the graph in (E) indicate WASP band intensity divided by the actin band intensity, and resulting ratios normalized to the values at the 0′ time point (non-activated cells), from three independent experiments (*n* = 3). *P* value * = 0.025, using paired two-tailed *t*-test.

F–H    Actin dynamics in mature synapse of LifeAct-GFP-expressing T cell using TIRFM. The images show the foci (pseudocolored red) extracted from the total synaptic F-actin (F, G), and kymograph shows the time course of foci and lamellar activity over a period of 210 s. Histograms of foci and lamellar protrusion and retraction (fluctuations) dynamics in WT T cell (H); lifetime of individual foci and lamellar events (graph on the bottom right, each dot represents a single foci/lamellar protrusion–retraction event *n* = 126 for foci and *n* = 77 for lamellar protrusions). ***P* < 0.0001 using Mann–Whitney two-tailed test between populations of cells within the same experiment.

I    Kymograph of the lamellar activity of WASP-deficient cells that lack actin foci.

Data information: Central values in the graphs represent Mean, and error bars represent ± SEM. These experiments were repeated at least thrice with similar results. Scale bar, 5 μm.

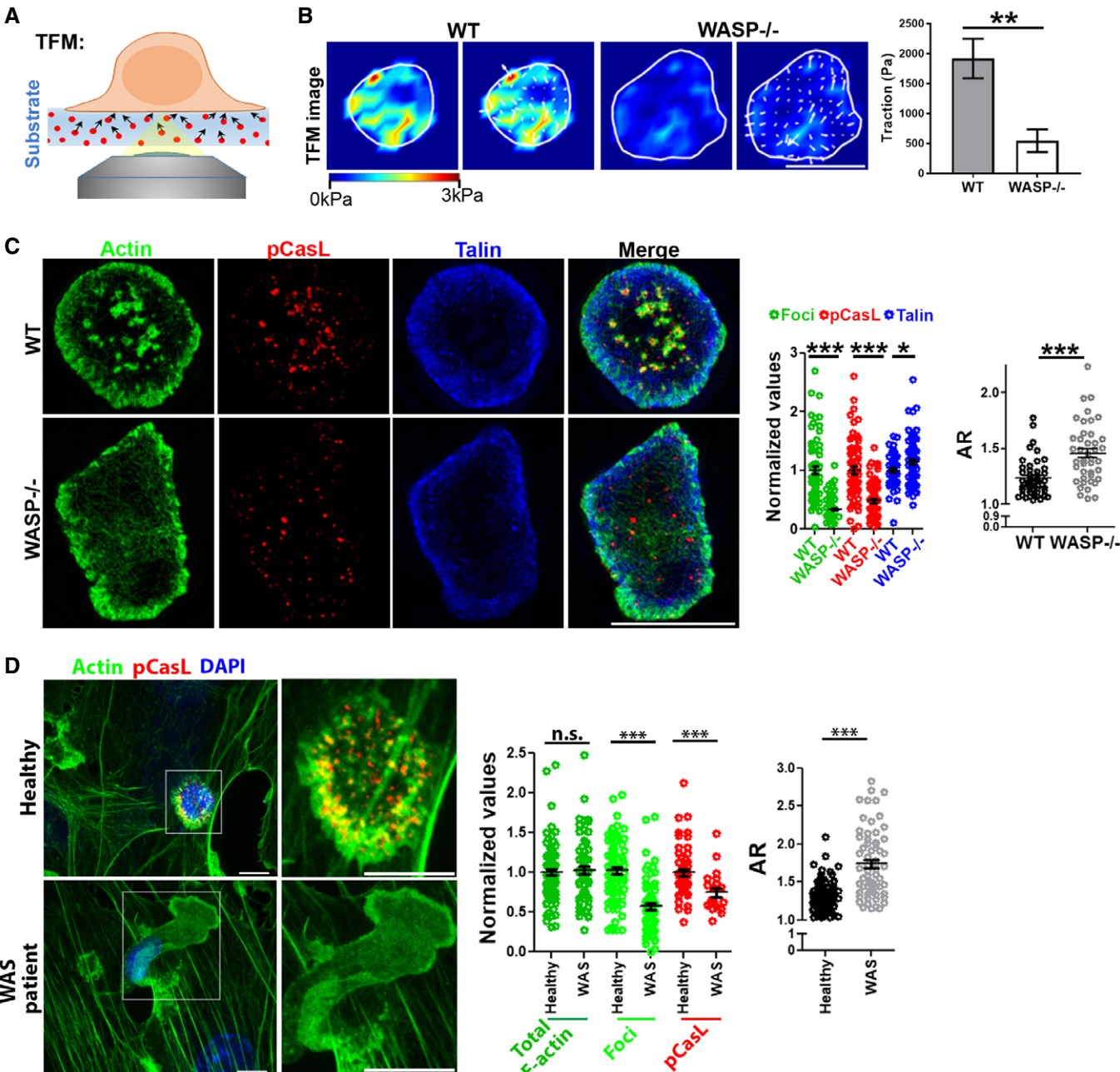

**Figure 3. Foci-associated mechanical forces are linked to synapse symmetry.**

A, B    Actin foci-deficient cells display poor traction forces in their synapse. WT or WASP$^{-/-}$ T cells were incubated on polyacrylamide substrates covalently functionalized with anti-CD3 and ICAM1, and traction force measurements were carried out as described in "Materials and Methods". The images in the right show traction force maps without (left panels) or with (right panels) force vectors. P value, **P < 0.005 as determined by using Mann–Whitney two-tailed test between populations of cells within the same experiment, n = 9 and 11.

C       SIM imaging of WT or WASP$^{-/-}$ T cells activated by APS for 5′. The graphs show quantification of actin foci (derived from phalloidin staining), pCasL, and talin levels in the synapse normalized to the "WT" mean value in each case (Middle graph; in the case of foci, n = 63 for WT and 86 for WASP$^{-/-}$; in the case of pCasL n = 63 for WT and 78 for WASP$^{-/-}$; in the case of talin, n = 59 for WT and 73 for WASP$^{-/-}$; in the case of AR, n = 46 for WT and 42 for WASP$^{-/-}$); right graph shows AR measurement. ***P < 0.0001; P value for talin *P = 0.07, as determined by using Mann–Whitney two-tailed test between populations of cells within the same experiment.

D       Human WAS patient CD4$^+$ T cells–APC conjugates display synapse symmetry defects similar to those observed in mouse WASP$^{-/-}$ CD4$^+$ T cells. CD4$^+$ T cells purified from healthy controls or WAS patients were incubated with superantigen-loaded HUVEC cells for 5′ (APC; see "Materials and Methods") and imaged using confocal microscopy. Each image represents a maximum intensity projection of the synaptic area. Graph in the middle shows values normalized to the mean value of the "Healthy" case. In the case of "actin" and "actin foci", n = 87 for healthy and n = 63 for WAS; in the case of pCasL, n = 56 for healthy and n = 21 for WAS; and for "AR" graph, n = 88 for healthy and n = 63 for WAS case. P values; ***P < 0.001; n.s. P = 0.09, as determined by using Mann–Whitney two-tailed test between populations of cells within the same experiment.

Data information: Central values in the graphs represent Mean, and error bars represent ± SEM. These experiments were repeated at least twice with similar results. Scale bar, 5 μm.

live antigen-presenting cells (APCs). Similar to the murine WASP$^{-/-}$ cells activated on BMDCs (Appendix Fig S8A), WAS cells showed poor mechanotransduction at early synapse, whether activated using APS (Appendix Fig S8B) or APCs (HUVECs, Fig 3D), and showed more polarized synapses than WT T cells. Although total F-actin content was normal in WAS synapses, there was a specific reduction in the number of actin foci. Faster synapse symmetry breaking in WAS cells was not due to a developmental defect, since transient reduction in WASP levels in healthy human T cells using short-hairpin RNAs (shRNAs; Kumari *et al*, 2015) yielded similar defects in pCasL and symmetry as seen with WAS cells (Fig EV5A). Furthermore, the reduced synaptic pCasL in these cells was specifically due to the loss of the actin nucleation-promoting factor (NPF) activity of WASP. Overexpression of a dominant negative WASP construct defective in NPF activity caused deficits in foci and pCasL that were indistinguishable from those seen with total WASP deficiency (Fig EV5B). Together, these data indicate that lack of nucleation of actin foci underlies premature symmetry breaking in WASP$^{-/-}$ T cells.

### In-plane cytoskeletal tension bolstered by actin foci dynamics promotes synapse stability

TFM measurements and pCasL accumulation showed that actin foci are associated with mechanical forces at the synapse. However, traction forces themselves do not explain how foci could promote synapse symmetry, since traction alone does not lead to interface symmetry and could instead be utilized to drive contact breaking and cell migration (Parsons *et al*, 2010). To explore how foci-associated cytoskeletal forces could generate a symmetric synaptic interface, we utilized a computational model of the actin cytoskeletal architecture and synaptic interface mechanics. This model was based on the distinct dynamics of the two major cytoskeletal networks observed at the synapse—actin foci and the peripheral lamella (Fig 2G and H). The model incorporated three key differences observed between these actin networks: (i) The foci are sites with higher actin nucleation than the lamella, as measured in Kumari *et al* (2015), (ii) the foci are sites where F-actin could be immobilized with a lower unbinding rate than at the lamellar sites given the longer lifetime of foci than the lamellar fluctuations observed in Fig 2H, and (iii) foci are situated across the synapse compared to the lamella which is only peripheral (Fig 4A). We employed a Brownian dynamics model of actin dynamics at the cell–substrate interface (Kim, 2015; Mak *et al*, 2016). The T-cell contact interface was approximated as an 8 μm square, and a thin volume of the cytoplasm just inside the plasma membrane at the contact was modeled (500 nm thick), mimicking the submembrane actin-rich region (Fritzsche *et al*, 2013; Linsmeier *et al*, 2016; Fig 4A). Actin foci were modeled as a regular array of 400 nm square sites across the contact plane (small orange squares in Fig 4A), and the peripheral lamella was modeled as a 400 nm wide zone at each edge of the contact (green). Actin nucleation and actin-linked receptor adhesions to substrate ligands could occur in all of these zones (Fig 4A). In this framework, F-actin evolves through a series of reactions comprising basic elements of actin dynamics: (i) polymerization/depolymerization/nucleation of actin filaments, (ii) actin crosslinking proteins linking/unlinking actin fibers, (iii) binding and unbinding of actin filaments to adhesion sites on the

substrate and (iv) myosin-mediated contractile forces applied to the cytoskeletal network, with actin nucleation and unbinding rates that differ between the indicated zones (Fig 4A).

Simulations of synaptic actin dynamics, cytoskeletal network morphology, and the resultant network tension profile over time were examined under conditions when actin foci are present (modeling wild-type cells; "WT") or absent (modeling WASP$^{-/-}$ cells). In WT cells, these simulations predicted that actin foci act as nodes supporting an interconnected actin network across the synaptic interface (Fig 4B left panels, Movies EV6 and EV7). This foci-connected actin network generates and sustains tension across the interface for prolonged durations (Fig 4C; synaptic stress is a cross-sectional area-normalized measure of the cytoskeletal network tension, parallel to the cell–substrate interface; see "Materials and Methods"), fueled by continual nucleation of filaments at the foci regions and associated myosin contractility (Appendix Fig S9A, Movies EV6 and EV7). If simulated using a curved surface of instead of a flat surface, the interface still showed foci interconnected with high stress fibers indicating that curvature of the APC surface may not affect the foci-dependent tension generation mechanism (Appendix Fig S9B). In the absence of actin foci (WASP$^{-/-}$), the F-actin network rapidly clears from the central areas of the "synapse" and assumes a lamella-like distribution (Fig 4B; Movie EV8). As a result, myosin II and, as a consequence, cytoskeletal stresses are enriched in the periphery accompanied by a relaxation in overall synaptic cytoskeletal tension (Fig 4B and C). To model the late stages of the T-cell synapse, we first allowed a foci-rich network to form and then downregulated WASP by eliminating nucleation of filaments at foci zones. In this scenario, the model predicts an instantaneous dissolution of foci and rapid redistribution of the F-actin/myosin II network to peripheral lamellar zones, causing reduction in overall synaptic cytoskeletal tension ("WT transitional" Fig 4B and C, Movie EV9). The reduced cytoskeletal tension in WASP$^{-/-}$ synapse predicted by the model would translate into poor traction stresses on the substrate. Indeed, we had observed poor traction stresses in WASP$^{-/-}$ synapses compared to the WT synapses in our traction force measurements (TFM; Fig 3B).

The differences in F-actin network architecture between stable and unstable synapses predicted by our simulations were validated by super-resolution imaging of F-actin organization at the synapse of live WT T cells that were transitioning from arrested to polarized states. Inter-foci connections were visible at the stable synapse of a T cell expressing LifeAct-GFP (Fig 4D—"0 s"). Both foci and these actin interconnections were lost as the cells polarized and regained motility (Insets, Fig 4D). These results suggest that a restraining tension within the actomyosin network is sustained and supported by actin foci across the contact area. Without foci, F-actin is depleted from the central zone and largely enriched in the peripheral lamella, along with myosin activity, as is expected in adherent cells poised to break symmetry (Yam *et al*, 2007; Lomakin *et al*, 2015). Indeed, ultrastructural visualization of the synaptic cytoskeleton using platinum replica electron microscopy (Heuser & Kirschner, 1980; Svitkina, 2009) revealed a tense network interconnected with "globular" nodes in the WT T cells (Fig 4E, left image, inset). WASP$^{-/-}$ cell micrographs showed a lack of foci and prevalence of loose "wave-like" clusters of filaments in the lamella resembling lamellipodial protrusions (Fig 4E, right image, inset; Svitkina, 2009). It is important to note that while it is the persistent lamellar/

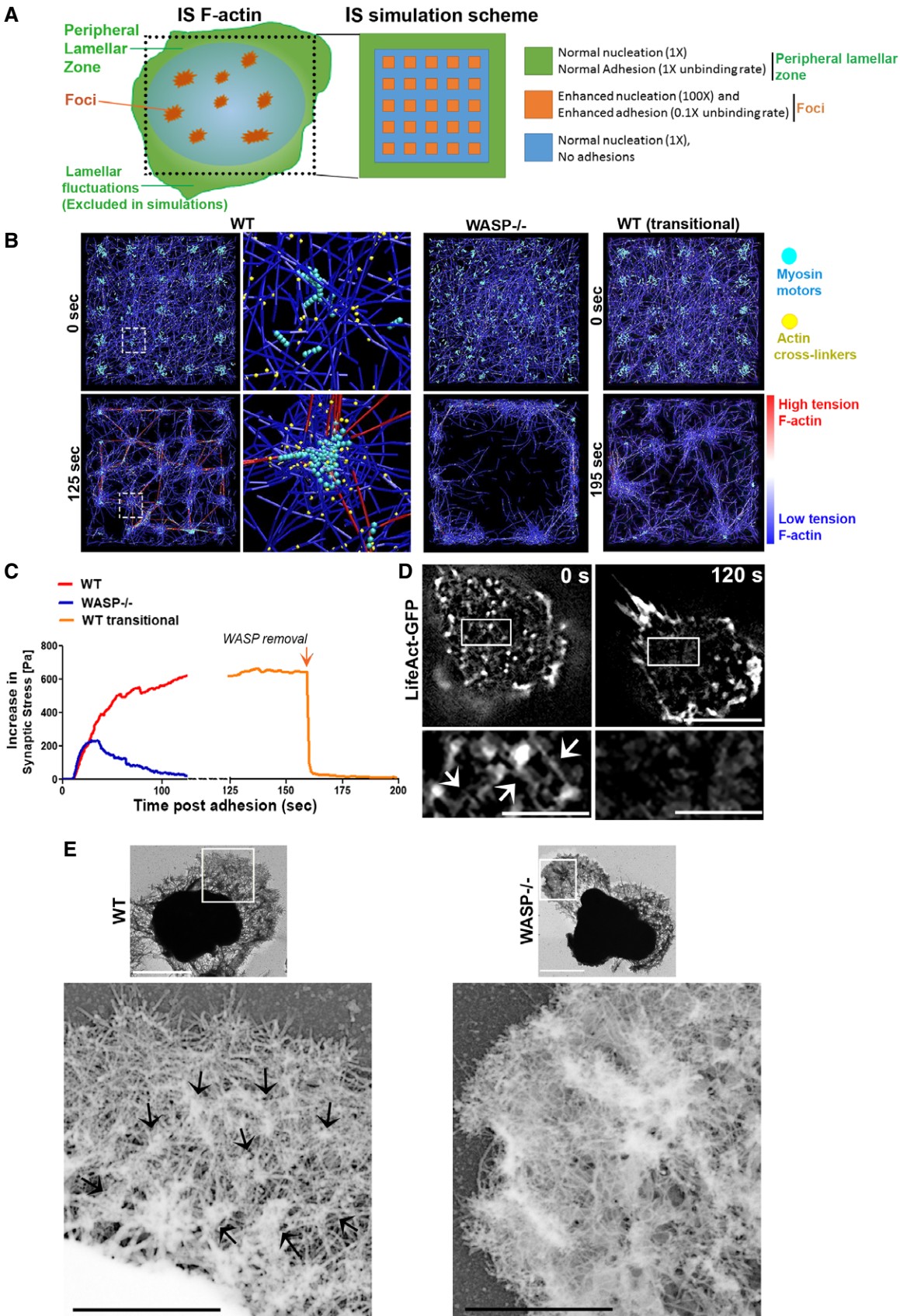

**Figure 4.**

◀

**Figure 4. Computational simulations to explain the cytoskeletal basis of symmetry sustenance at synapse.**

A   Outline of the spatial scheme utilized to simulate synapse cytoskeletal dynamics. Cartoon on the left shows the spatial patterning of the two major actin networks: the lamellar F-actin network in the peripheral zone that would continuously undergo fluctuations, and foci. The foci are interspersed throughout the synapse within a dynamic lamellar F-actin network. The peripheral lamellar fluctuations seen in live imaging could not be included in the simulations due to unavailability of data on fresh boundary adhesions. The cartoon on the right shows square simulation space patterned after the cartoon on the left and utilized here for simulations. The actin nucleation and adhesion parameters used at the foci and lamellar zones are indicated on the right.

B   Simulation of T cell IS F-actin behavior and resultant mechanical tension incorporating the differential dynamics and positioning of foci and lamella across the synaptic interface, using Brownian dynamics equation with no inertia (see "Materials and Methods"). The images show simulation snapshots at the beginning (0 min; soon after the attachment and spreading of T cells on the substrates) and the end of the simulations for the three IS cases—with persistent foci (WT, left most panels), with no foci (WASP$^{-/-}$, middle right panels), and with WASP and consequently foci downregulated after a period of synapse maturation (WT transitional). The colors in the images indicate myosin II motor (pseudocolored teal), F-actin crosslinking proteins (pseudocolored yellow), and the F-actin filaments (pseudocolored blue for the low tension and the red for thigh tension actin filaments); middle left image panels show the distribution of the aforementioned proteins at individual foci site at a higher magnification.

C   Predicted stress (tension normalized to the area) profiles within the cytoskeletal architecture as the synaptic contact progresses in time, corresponding to the three cases shown in (D).

D   Live imaging of a T cell using SIM shows the inter-foci connections (arrows in the inset) as predicted in the simulations, and their loss concomitant with loss of actin foci, as the T cell polarizes to initiate motility (compare insets between 0 and 120 s). "0 s" refers to the beginning of the observation of the cell, after it has attached to the substrate, spread, and maintained synapse for 12 min. Scale bar, 5 μm in main images and 1.5 μm in the magnified insets.

E   Ultrastructural visualization of detergent extracted WT and WASP$^{-/-}$ T-cell synaptic cytoskeleton using platinum replica electron microscopy (top images). Each image shows individual synapse, and dark zone in the middle of the synapse represents cell nucleus. Scale bar, 2.5 μm. The bottom left micrograph is the magnified inset from WT synapse and shows actin foci (identified as globular cluster of short filaments in the inset, marked with arrows)-dependent interconnected cytoskeletal architecture that is associated with synapse symmetry as predicted in the model, and is lacking in the WASP$^{-/-}$ cell synapse (bottom right micrograph-magnified inset from WASP$^{-/-}$ synapse). Scale bars, 1 μm. This experiment was repeated at least twice with similar results.

lamellipodial protrusions that might eventually drive motility when symmetry is broken and synaptic stress is low (Yam *et al*, 2007; Lomakin *et al*, 2015), we could not model the protrusions in our simulations since experimentally measuring boundary adhesion dynamics in primary T cells is technically challenging as a results of their small size (~ 5 μm diameter) and rapid cortical dynamics.

**Synaptic tension required for synapse stability is generated by an interplay of actin foci and myosin II activity**

In the computational model, myosin II contractile activity plays a major role in generating cytoskeletal network tension across the synapse. Since local intracellular cytoskeletal tension dynamics are difficult to measure experimentally, we instead tested the role of myosin-mediated tension generation via acute pharmacological perturbations of myosin II to uncouple actin foci from myosin-mediated tension generation. Inhibition of myosin II using blebbistatin led to reduced pCasL levels and premature T-cell polarization, even though foci and talin accumulation at the interface were not reduced (Fig 5A–C). Thus, myosin II functions downstream of foci to generate cytoskeletal contractile tension, and loss of this mechanical tension promotes T-cell polarization. pCasL levels in blebbistatin-treated cells were not as low as in WASP$^{-/-}$ cells, which may reflect residual cytoskeletal stresses, potentially protrusive, arising from actin polymerization at the actin foci (Pollard & Borisy, 2003; Kumari *et al*, 2015). Blebbistatin treatment also did not further exacerbate the tendency of WASP$^{-/-}$ T cells to break symmetry, as predicted by our model (Fig 5D and E). Cells treated with the Arp2/3 complex inhibitor CK666 that preferentially ablates foci (Kumari *et al*, 2015) result in increased AR, demonstrating that pharmacological inhibition of actin foci results in reduced pCasL and an increased tendency for T-cell polarization (Fig 5B–E, Movie EV10).

To further examine whether the synapse-wide transmission of foci-dependent cytoskeletal tension is important for maintenance of a symmetric synapse, we carried out simulations of the actin network as in Fig 3, in which we allowed the actin network to evolve for 120 s and then selectively ablated F-actin network

connectivity and tension in a spatially controlled manner (dashed area in the simulation snapshot of Fig 5F). This was achieved using localized myosin II inactivation and crosslinking of associated F-actin, and this perturbation was found to cause a disruption in synaptic F-actin organization and symmetry, as well as a significant lowering of overall synaptic tension (Fig 5F and G, Movie EV11). To experimentally test the prediction that the symmetry of F-actin tension is important for synapse maintenance, we utilized a photoreactive form of blebbistatin, azido-blebbistatin, to optically inhibit myosin II and thus freeze the actomyosin network in a spatially controlled manner (Kepiro *et al*, 2012). Azido-blebbistatin binds to myosin II and, following excitation with UV light, crosslinks with myosin II molecules, effectively freezing actomyosin contractions in a spatially controlled fashion. LifeAct-GFP-expressing T cells forming a symmetric synapse were treated with low doses of azido-blebbistatin, photoactivated locally on one edge of the cell (dashed boxes in Fig 5H) and imaged under low excitation power to minimize phototoxicity. Photoactivation of the inhibitor induced an immediate deflection of the cellular center of mass (COM) and transition to a polarized morphology (Fig 5H and I, Movie EV12). The COM deflection after photoactivation was specific to the localized myosin II perturbation and not due to global inhibition of myosin II by free azido-blebbistatin in the culture media, since without photoactivation inhibitor-treated cells were indistinguishable from the DMSO-treated control cells (Appendix Fig S10).

Continual actin nucleation at actin foci can generate membrane protrusions into the substrate. To examine whether synapse symmetry is influenced by the protrusive behavior of the foci, in addition to the intracellular myosin-based in-plane tensional mechanism described above, we activated T cells on substrates of varying rigidities. We found that actin foci supported synapse symmetry regardless of the substrate stiffness (Fig 6A–C), such that the symmetry responses seen previously on glass coverslips and physiologic APC were also seen on supported lipid bilayers (Fig 6A). The organization of F-actin into foci was critical for synapse symmetry maintenance, and total synaptic levels of ICAM1, talin, or F-actin did not alter with symmetry breaking (Fig 6B). A higher propensity to break

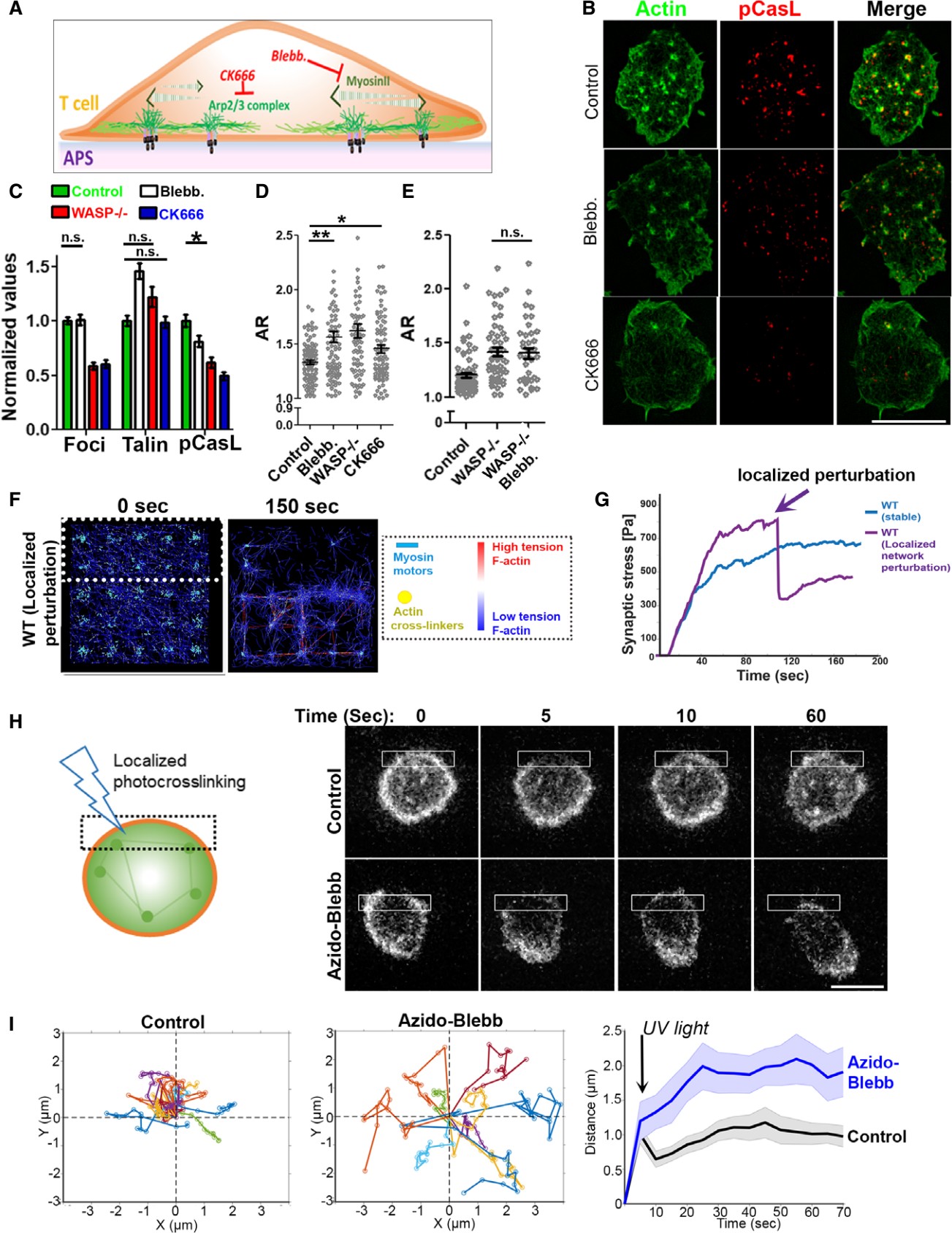

**Figure 5.**

**Figure 5.  Pharmacological interrogation of actin foci–myosin crosstalk in sustaining symmetry.**

A       Schematic of the cytoskeletal inhibitors and their targets used in the experiments.

B–E     SIM imaging of mouse T cells activated for 5′ on APS, then treated with the indicated inhibitors or vehicle control. Shown are the representative SIM images (B), and quantification of actin foci (from the phalloidin images), talin, or pCasL in the contact interface (normalized to the mean value in the control case, C), and measured cell morphology aspect ratios (D, E). Since CK666 treatment reduces F-actin intensity (Kumari *et al*, 2015), the "actin" image of the CK666-treated cell in (B) was contrasted differently than the "control" and "blebb." cases for a clearer visualization of F-actin. *P* values, *$P$ = 0.02, **$P$ = 0.001, n.s. > 0.05. For the rest of the comparisons between the control and treatment cases, $P < 0.0001$. For the bar graph in (B), *n* for foci groups are 214, 83, 92, and 98; for pCasL groups are 100, 68, 91, and 92; and for talin groups are 102, 70, 89, and 93, respectively. *P* value for the n.s. cases is > 0.05, *$P$ = 0.02. In the foci, pCasL, and talin groups, when not indicated, the *P* values are < 0.0001 using Mann–Whitney two-tailed test. Central values in the graphs represent Mean, and error bars represent ± SEM. These experiments were repeated at least thrice with similar results. Scale bar, 5 µm.

F, G    Simulation of synaptic F-actin network following localized myosin inactivation and inactivated myosin-mediated F-actin crosslinking within the area marked with dotted line in the left image. The graph shows the predicted alteration in mechanical stress profile, as myosin is inactivated crosslinked in the defined region (marked by an arrow), (G).

H, I    Experimental confirmation of localized actomyosin crosslinking and symmetry breaking. LifeAct-GFP-expressing T cells were incubated with activating substrates and treated with vehicle control or 1 µM azido-blebb and their contact interface imaged using low excitation laser power (0.5 mW), using a spinning disk confocal microscope (H). The montage shows the shape change and movement that synapse undergoes after it is exposed to a brief flash of blue light crosslinking azido-blebb and actomyosin network within the illuminated area (shown with a box in the image). The deflection in center of mass (COM) within a minute was calculated and plotted for at least 10 cells in each case (graphs in I). The left graphs in (I) show individual COM trajectory, and the graph on the right shows average COM movement of cells within a minute post-photo-inactivation of myosin II. This experiment was repeated twice with similar results. Scale bar, 5 µm.

the synapse was also observed in WASP$^{-/-}$ cells activated on hydrogels of low (2.5 KPa) or high (100 KPa) stiffness (Fig 6C).

Thus, symmetry maintenance requires foci nucleation and myosin contractile activity across the synaptic interface, and this mechanism is robust to varying levels of substrate stiffness. The perturbation of this tensional mechanism via loss of foci, myosin II inhibition, or localized actomyosin crosslinking leads to a loss of cellular symmetry.

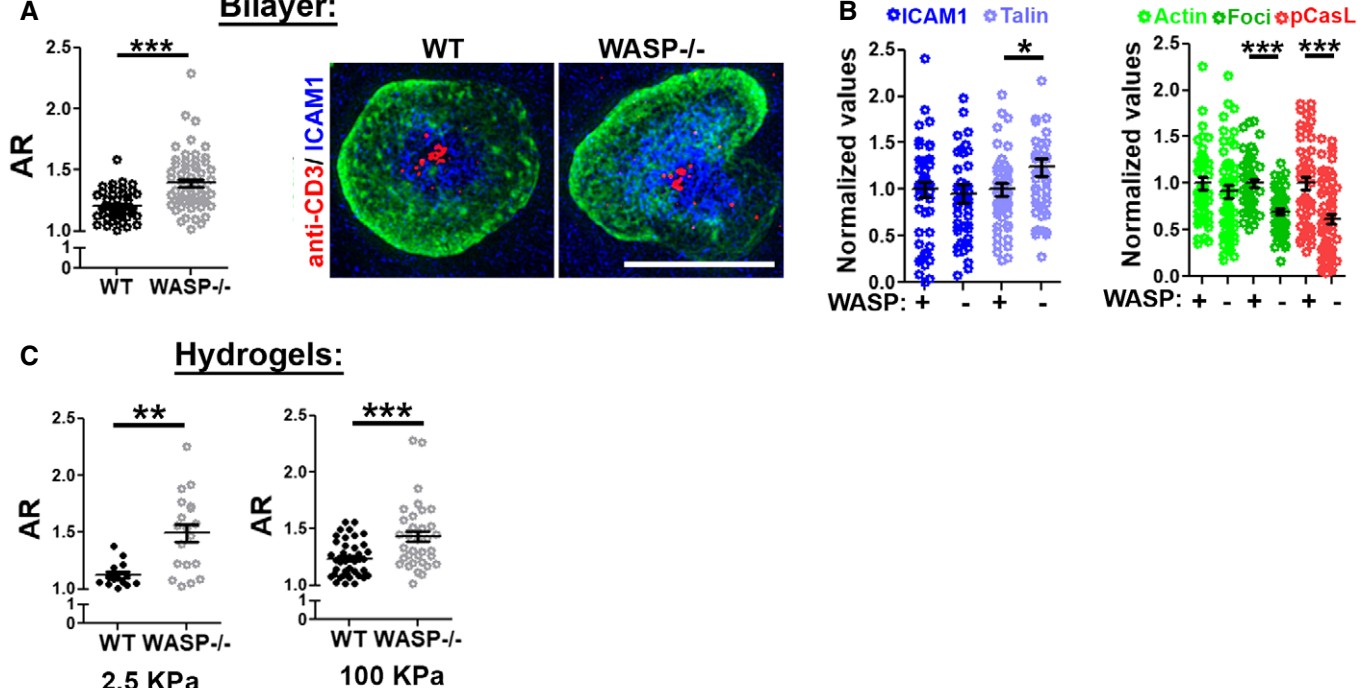

# Discussion

Rapid transitions between distinct motility states underlie T-cell immune function. The motile state of T cells is associated with immunosurveillance, while antigen-induced arrest is considered to be a hallmark of activation, both *in vitro* and *in vivo* (Hugues *et al*, 2004; Mempel *et al*, 2004; Celli *et al*, 2007; Skokos *et al*, 2007; Zaretsky *et al*, 2017). Here, using naïve primary T cells we have identified and characterized an antigen-induced cytoskeletal tension program that actively prolongs their arrested or "synaptic" state. The experiments presented here reveal some surprising facets of lymphocyte synapse. Unlike most other cellular adhesion systems, the T-cell synapse is sustained independently of integrin activation. In addition, synapse stability is achieved regardless of the substrate stiffness. These results imply that that T cells could tune their synapse longevity using the cytoskeletal mechanism described here, regardless of the stiffness of the partner antigen-presenting cells. These results presented in this study also provide a conceptual framework to assess primary T-cell synaptic actin organizational defects in observed in WAS and other actin-related primary immunodeficiency diseases.

Symmetry breaking and motility progression using lamellar protrusions are fundamental features of T-cell behavior since it is required for effective immunosurveillance. Our study suggests that maintaining a stable synapse for prolonged periods requires the active formation of actin foci by WASP to stabilize the synapse against lamellar fluctuations. These fluctuations are a constant and ongoing phenomenon in the T-cell cortex (Sims *et al*, 2007; Roybal *et al*, 2013; Krummel *et al*, 2014; Ritter *et al*, 2015) and are also seen during the synaptic phase. Consequently, our data indicate that the synapse would destabilize when WASP is degraded and new actin foci ceased to form. The clock for WASP/foci degradation and subsequent symmetry breaking is likely set by the cellular machinery that degrades WASP, perhaps in sync with the degradation of TCR signaling components. In line with this hypothesis, previous studies have described a global protein degradation program that is triggered by T-cell receptor activation (Heissmeyer *et al*, 2004; Wiedemann *et al*, 2005; O'Leary *et al*, 2015). WASP is a substrate of this program (Macpherson *et al*, 2012; Reicher *et al*, 2012; Watanabe *et al*, 2013), indicating that WASP degradation may result from the ubiquitination and proteolysis programs that mediate downregulation of TCR signaling.

The results presented in this study indicate that the mechanisms of synapse formation and synapse maintenance are mechanistically distinguishable. While increased F-actin is required for the former, the latter depends on changes in F-actin organization rather than F-actin levels. Similarly, activation of the integrin LFA1 is a prerequisite for successful synaptic engagement but is not needed for maintenance of the synapse. These results support a recent finding that the LFA1-ICAM1 interaction is not necessary for stable CD4$^+$ T cell–DC synapses in lymph nodes (Feigelson *et al*, 2018). The physiological implications are that T cells may use integrin-based adhesion to initiate synaptic contacts with selective cellular partners (Zaretsky *et al*, 2017; Feigelson *et al*, 2018), but then maintain adhesion using lateral tension within the synapse driven by dynamic cytoskeletal mechanisms comprising foci nucleation and actomyosin contractions. Our data also show that TCR microclusters can migrate independently of WASP in T cells activated on supported lipid bilayers. These data

indicate that either a WASP-independent pool of F-actin such as the filaments nucleated via formins (Murugesan *et al*, 2016) or microtubule motor dynein (Hashimoto-Tane *et al*, 2011) could transport TCR microclusters in WASP$^{-/-}$ cells (Colin-York *et al*, 2019), since both have been shown to be a mode of transport for TCR microclusters at the synapse.

The immediate arrest of T cells when they encounter APCs displaying high-affinity antigens and the concomitant formation of a symmetric contact has been hypothesized to be driven by a "stop signal" (Dustin *et al*, 1997; Dustin, 2002, 2008b; Hugues *et al*, 2004; Mempel *et al*, 2004; Downey *et al*, 2008). We find that actin foci are an integral part of the cellular machinery associated with maintaining TCR-induced motility arrest. Previous studies have linked intracellular calcium flux in T cells with the stop signal (Negulescu *et al*, 1996; Dustin *et al*, 1997). While it is possible that actin foci machinery could communicate with the calcium stop-signal pathway for coordinating synapse stability, it is unclear how this communication might take place. At the cellular level, we found evidence that the calcium pathway depends on WASP-dependent actin foci to promote synapse stability. A thapsigargin-induced rise in intracellular calcium could not arrest the WASP$^{-/-}$ T cells. In contrast, we show that the foci-mediated synapse stabilization is independent of intracellular calcium because BAPTA treatment did not reduce foci or promote synapse symmetry breaking. Detachment of T cells from APCs was shown to be associated with a period of reduced TCR signaling (Bohineust *et al*, 2018). Whether a reduction in calcium signaling during the transition from stable to motile state might underlie antigen receptor unresponsiveness remains to be investigated. It is important to note that the pharmacological calcium elevation we used to examine the relationship between foci and intracellular calcium can alter some basic features of the synapse, such as downregulation of talin, and generate minor off-target effects on contact symmetry and actin foci. Furthermore, a previous study proposed that T cells could selectively employ either actin-based or calcium-based mechanisms to promote synapse stability based on antigen affinity (Moreau *et al*, 2015). This implies that the calcium or foci-mediated stop-signal pathways could be selectively dominant depending on the strength of the TCR–antigen interaction.

At the functional and architectural level, actin foci represent a structural intermediate between podosomes and focal adhesions (FAs) that have been described in innate immune and adherent cells, respectively. Continual actin nucleation at foci as well as some of their molecular and organizational characteristics resembles those of podosomes observed in innate immune cells (Cervero *et al*, 2018). Both are WASP-dependent, enriched in cortactin homolog protein HS1, maintained by continuous actin nucleation, associated with contact symmetry breaking in macrophages (Cervero *et al*, 2018), and are capable of generating protrusions. However, several key features distinguish the two. Actin foci are antigen receptor-triggered (Kumari *et al*, 2015) and display a lifetime at least an order of magnitude shorter than podosomes. Podosomes have a polarized distribution within macrophages during symmetry breaking of the contact interface (Cervero *et al*, 2018), whereas foci are downregulated in primary T cells during synapse breaking. Additionally, protrusive behavior of the podosomes is crucial for cell adhesion. Foci-dependent synapse stability proceeds regardless of the stiffness of the substrate indicating that the protrusion of foci into the substrate may not be required for synapse stability. Additionally,

our computational simulations predict that foci could instead pull on the substrate via membrane receptors. Although we have not confirmed the identity of these receptors, they are likely to include TCR microclusters. With regard to their mechanism of synapse stabilization by generating an actomyosin system that organizes and scales mechanical tension across the cell–substrate contact interface, foci resemble the FAs in fibroblasts (Kanchanawong *et al*, 2010). However, there is a key difference between the foci and FA-based adhesion mechanisms as well. FAs attach to stable actin filaments. If a cell can generate high enough tensile forces, it can rupture these FA-based attachments and detach from the substrate. In contrast, foci continuously nucleate and generate actin filaments and would dissipate local tensile forces preventing cellular detachment. Therefore, the only way to overcome the foci-based adhesion is to down-regulate their formation, which is what we see in cells that are preparing to migrate. In this regard, actin foci are a departure from the classical "clutch mechanism" proposed for the FAs (Kanchana-wong *et al*, 2010) and represent an adhesion modality that T cells have evolved for their specialized motility requirements.

Previous studies have provided evidence for existence actin architecture resembling the actin foci and actomyosin cables described in this study. For example, a recent study reported the existence of actin "vortex-like foci" that are interconnected by the actomyosin network in NK cells (Fritzsche *et al*, 2017; Carisey *et al*, 2018). It remains unclear whether these cytoskeletal structures reported previously and the foci described here involve similar molecular mechanisms for formation or have related functions in synapse sustenance.

Throughout their lifetime, T cells encounter antigens in diverse mechanical microenvironments. The antigen-induced tensional pathway identified here could represent a T cell-intrinsic tool to accommodate mechanical disparities between antigen-presenting cells and achieve a stable synapse in order to maintain optimal antigen sampling time. Different T-cell subtypes are known to display varying motilities even when they encounter the same APC type (Tadokoro *et al*, 2006), and T cells encounter antigens of varying affinities for their TCRs. How the actin foci-based mechanisms outlined here operate in these diverse contexts remains an open question. Simultaneous examination of the effects of antigen affinity on foci, TCR triggering, and T-cell activation in different T-cell classes will help establish a relationship between TCR–antigen affinity and mechanotransduction via this tensional mechanism at the immunological synapse. Future studies will also be needed to clarify the ways in which the described mechanical deficits in the synapses formed by WASP-deficient T cells contribute to the WAS immunodeficiency at the whole-organism level (Thrasher & Burns, 2010).

# Materials and Methods

## Cells and reagents

For most experiments, unless otherwise mentioned, mouse primary CD4$^+$ T cells were isolated from C57BL6/j WT or WASP$^{-/-}$ Mice lymph organs, using EasySep™ Mouse CD4$^+$ T Cell Isolation Kit (StemCell). Cells were rested at 37°C for few hours in culture media (RPMI + 10% FBS + 100 μM β-merceptoethanol + 1 mM sodium pyruvate) and used for experiments thereafter. For live imaging, cells were cultured in phenol red free XVivo-10 media (Lonza) supplemented with 10 units/ml IL-2. For antigen-specific CD4$^+$ T cells, CD4$^+$ T cells were isolated from OTII WT or WASP$^{-/-}$ animals, using the procedure as described above.

For isolating human CD4$^+$ T cells, human peripheral blood from healthy donors was acquired from the New York Blood Bank or from WAS patients at Boston Children's hospital (two patients, males). The blood samples were transported, handled, and processed in accordance with the OSHA guidelines. The cells were isolated from the blood using the CD4$^+$ rosette sep. kit, rested at 37°C in culture media for a few hours and were used in the experiments thereafter.

Unless otherwise mentioned, the inhibitors were purchased from Sigma chemicals and were used at a concentration indicated in the figure legends. The antibodies for immunostaining including phopho-Y249Cas (#4014), phospho-Y171 Lat (#3581), phospho-Y397HS1 (clone D12C1), and phosphor-Y139/SykY352 Zap70 (clone 65E4) were purchased from Cell Signaling Technology. Anti-phospho-Y145 (#EP2853Y, Novus), anti-tubulin (F2168, Sigma), anti-MyosinII (PRB440P, Covance), anti-WASP (Chicken polyclonal for immunostaining, #Gw22608, Sigma), anti-WASP (Rabbit polyclonal for immunostaining, #SAB4503087, Sigma), anti-phospho Y290 WASP (#Ab59278, Abcam), and anti-talin (clone C20, Santa-cruz) were purchased from the indicated vendors.

## Cell activation substrates

In most experiments, glass coverslip (#1.5 chambered coverglass, Thermo Scientific) coated with 10 μg/ml anti-CD3 and 1 μg/ml ICAM1 was used as activating substrate. For cell–cell conjugate experiments, either IFNγ (100 units/ml, overnight)-treated and superantigen-loaded (TSST-1 + SEB cocktail, 1 μg/ml for 1 h at 37°C) HUVEC cells (Kumari *et al*, 2015) or antigen-loaded, differentiated primary BMDCs were used for synapse assays with human or mouse T cells, respectively. In experiments examining the distribution of ICAM1 on activating substrates, or measuring synapse stability (AR) on supported lipid bilayers, planar lipid bilayer system was used. Glass-supported lipid bilayers were deposited and were reconstituted with anti-CD3-Alexa568, and ICAM1X12-his- cy5, and incubated with T cells as described previously (Kumari *et al*, 2015).

## Microscopy

### Interference reflection microscopy

For IRM, LabtekII chambers were coated with anti-CD3 and ICAM1, were washed, and were incubated with the CD4$^+$ T cells suspended in the growth media. Care was taken to maintain the temperature of the T cells to 37°C, between the transfers from the culture dish to imaging chambers. The chambers were imaged using a temperature-controlled 37°C stage adaptor on a Zeiss LSM510 confocal microscope equipped with a 63× 1.40 NA objective. The samples were imaged using the 543-nm laser at 0.1 mW and collected using Zeiss LSM software. Images were analyzed using Fiji Software.

### Structured illumination microscopy

For SI super-resolution microscopy, fixed cells were imaged with an OMX-3D microscope, V3 type (Applied Precision, GE), equipped with 405-, 488-, and 594-nm lasers and three Photometrics Cascade

II EMCCD cameras. Images were acquired with a 100×, NA 1.4 oil objective at 0.125-μm Z steps using 1.518 immersion oil at room temperature. The images were acquired under the same illumination settings across the individual experiment and then processed with OMX SoftWoRx software (Applied Precision, GE). Images were saved in the TIFF format after reconstruction and alignment using optimized OTFs and wiener filter settings. The images were reconstructed using SIM SoftWoRx processing software and were then analyzed using Image J.

### TIRF
For the TIRF experiments, a Nikon revolution spinning disk confocal system equipped with a TIRF module and Andor iXON EMCCD camera was used. The images were acquired using 405-, 488-, 561-, and 642-nm lasers, a 100× objective (NA 1.49), and a 1.5× projection lens, and analyzed using Fiji software. The depth of TIRF field was maintained to 120–200 nm, as measured by optically resolved fluorescent beads.

### Confocal microscopy and azido-blebbistatin experiment
For imaging cell–cell conjugates, Nikon revolution spinning disk confocal microscope equipped with Yokagawa CSU-X1 module was used. Images were acquired using 100× objective (NA 1.49) along with a 1.5× projection and an Andor iXON EMCCD camera. For azido-blebbistatin treatment experiment, the abovementioned spinning disk system was used to acquire a 0.4 μm optical section in the plane of the synapse, coupled with live microscopy. Cells were allowed to form synapses and were treated with either inhibitor or DMSO just before imaging. During imaging, the inhibitor was activated by excitation with five brief pulses of 405-nm laser at 50 mW power employing the Andor FRAPPA photomanipulation system. Images were subsequently analyzed using Fiji software and plotted using MATLAB software.

### Lattice light sheet
The lattice light-sheet microscope (LLSM) used in these experiments is housed in the Advanced Imaged Center (AIC) at the Howard Hughes Medical Institute Janelia research campus. The system is configured and operated as previously described (Chen et al, 2014). Briefly, T cells were activated on 5 mm round glass coverslips (Warner Instruments, Catalog # CS-5R), plasma-treated and coated with anti-CD3 and ICAM1-extracellular domain, and imaged in culture media at 37°C. Samples were illuminated by lattice light sheet using 488-nm diode lasers (MPB Communications) through an excitation objective (Special Optics, 0.65 NA, 3.74-mm WD). Fluorescent emission was collected by detection objective (Nikon, CFI Apo LWD 25XW, 1.1 NA) and detected by a sCMOS camera (Hamamatsu Orca Flash 4.0 v2). Acquired data were de-skewed as previously described (Chen et al, 2014) and deconvolved using an iterative Richardson–Lucy algorithm. Point-spread functions for deconvolution were experimentally measured using 200-nm tetraspeck beads adhered to 5-mm glass coverslips (T7280, Invitrogen) for 488 excitation wavelength.

### Traction force measurements
Cell root-mean-square (RMS) tractions at the basal surface were quantified by measuring embedded fluorescent submicrometer particle displacement fields in the gel substrates, following published methods (Poh et al, 2012). Briefly, carboxylated red fluorescent submicron beads (0.2 μm in diameter; F-8810, Life Technologies) were embedded in the hydrogel substrates, the substrates were then covalently functionalized with anti-CD3 (2C11, 10 μg/ml) and ICAM-1-12XHis (2 μg/ml). Cells were seeded at a concentration of 2 million/ml and allowed to adhere at 37°C for 5 min before the traction force was measured. The un-adhered cells were removed by gentle aspiration with warm culture media before imaging the cells and beads. Fluorescence images of the microbeads were taken using Olympus FV1200 laser scanning confocal system equipped with 30× silicone oil objective (1.05NA), and 550-nm laser line, both when the cells were adhered, and again after cells were removed from the substrate using EDTA. Using a custom Matlab code, the displacement of the fluorescent beads at the apical surface of the hydrogel was computed. The RMS traction generated by the cell was calculated based on the displacement field and the known stiffness of the hydrogel substrate. The contractile moment in the synapse was analyzed by averaging the traction forces around two highest traction zones in the interface, and multiplying them by the distance separating them as described in Butler et al (2002). For all TFM experiments, substrate stiffness of the hydrogel was 100 kPa, prepared using published protocols (Engler et al, 2004; Yeung et al, 2005).

### Imaging cytoskeleton using platinum replica electron microscopy
Mouse naïve CD4$^+$ T cells were cultured on detergent (Hellmanex, Sigma #Z805939-1EA)-cleaned glass coverslips (12-mm round coverslips, TedPella #26020) coated with 10 μg/ml anti-CD3 and 1 μg/ml ICAM1 for 5 min at 37°C. The cells were then processed for platinum replica electron microscopy of cytoskeleton using a protocol described previously (Svitkina & Borisy, 1998). Briefly, cells were washed with pre-warmed PBS (three times), were then slowly incubated with extraction buffer (1% Triton X-100, 2% high molecular wt. PEG, 2 μM Phalloidin, 2 μM Taxol) for 5 min with occasional gentle shaking, and were then quickly washed with PEM buffer (100 mM PIPES, 1 mM MgCl$_2$, 1 mM EGTA) followed with incubation with fixative (2% in nacacodylate buffer, 0.1% tannic acid in DI water, 0.2% uric acid in water) for 45 min. Cells were then dehydrated with ethanol washes (50, 75, 90, 95, 100% X3 ETOH), critical point dried, rotary shadowed with platinum followed with carbon shadowing. The replicas were released using hydrofluoric acid onto EM grids and imaged using a Jeol 5600LV Scanning Electron Microscope. Images were further analyzed and negatively contrasted using ImageJ.

## Brownian dynamics model of active actin networks during T-cell activation

We simulated cytoskeletal networks that consist of actin filaments that can polymerize, depolymerize, and nucleate; actin crosslinking proteins (ACPs) that can bind and unbind in a force-sensitive manner; and active myosin II motors that walk along actin filaments generating tension. The model is based on Brownian dynamics and described in detail in our previous work with no inertia (Kim, 2015; Mak et al, 2016):

$$\boldsymbol{F}_i - \zeta_i \frac{\mathrm{d}\boldsymbol{r}_i}{\mathrm{d}t} + \boldsymbol{F}_i^T$$

where $\boldsymbol{F}_i$ is the deterministic force determined by the elasticity of the network; $\zeta_i$ is the drag of the medium; $\boldsymbol{F}_i^T$ is the stochastic thermal force; and $\boldsymbol{r}_i$ is the position of $i^{\text{th}}$ particle.

The equation is evolved via Euler's method and solved over time. Extensive details and parameter values, most of which are based on experimental data, of all features in our model can be found here (Mak *et al*, 2016). Table EV1 provides parameter values used in this study in order to investigate the spatial and temporal modulations of the actin cytoskeleton to mimic actin foci formation in the synapse during T-cell activation. Cytoskeletal network stresses (in the stress profile plots) are calculated by summing the normal component of tensional forces acting on cytoskeletal components across *x–z* and *y–z* cross-sections, divided by the cross-section areas.

To model the T-cell synapse formed against a flat surface, we simulated a thin 3D domain ($8 \times 8 \times 0.5$ μm) with active cytoskeletal components. In this domain, we defined regions with different local kinetics of dynamic cytoskeletal components. At the edges of the cell (400 nm width from each edge in the *x* and *y* directions), to mimic the integrin-rich peripheral lamella which is important in cell migration, adhesions between actin and the bottom surface are enabled. In the middle of the cell to mimic foci-mediated adhesions, a $5 \times 5$ grid of sites each $400 \times 400$ nm$^2$ enables adhesions between actin filaments and the bottom surface. No boundary adhesions are enabled in other parts of the domain. The nucleation rate of actin filaments is highest at the foci region. To mimic azido-blebbistatin treatment in our simulations, at 100s after the activation of myosin motor walking, we deactivated motor walking and actin-to-substrate adhesions at the top 40% of the domain.

Figure 4A shows the spatial profile of the kinetics of actin nucleation and adhesion used for simulations, and Table EV1 shows the parameter values used for the simulations. Because exact rates controlling actin dynamics in live cells are elusive due to the plethora of actin regulating factors (formins, Arp2/3, cofilin, capping proteins, etc.) and spatiotemporal biochemical signaling (e.g., via Rho GTPases), we chose rates of actin turnover comparable to experimental observations (Pollard, 2007) and probed idealized spatial profiles that can capture the spatial distributions of actin seen in T cells during activation.

### Image analysis and statistics

The images were analyzed using Fiji software. For determination of cell shape (AR), automated cell boundaries were creating by first thresholding the actin images to remove background, watershed plugin to fill the low intensity areas within the cell areas created by threshold operation, and finally with "analyze particles" utility of Fiji. Using the automated boundaries, the raw images were analyzed for the intensity of relevant proteins within each synapse. Foci were estimated as described previously (Kumari *et al*, 2015). Briefly, a moving window of $1.6 \times 1.6$ μm$^2$ was used to create a Gaussian blur image of the original raw images. The Gaussian blur image was subtracted from the raw image to create the foci image. Data were plotted using GraphPad Prism or MATLAB. For statistical analysis, Mann–Whitney non-parametric two-tailed test was performed to compare population of cells. In all graphs, unless otherwise mentioned, the dataset was normalized to the mean values of the "Control" population and then plotted as a "normalized" value. Most graphs highlight the Mean $\pm$ SEM in the scatter plot of normalized values, where each dot represents values obtained from a single cell, unless otherwise mentioned. All experiments were repeated thrice, unless mentioned otherwise in the figure legends.

## Data availability

Raw data associated with the figures will be made available on a reasonable request. The custom code used for the simulations will be made available on a reasonable request.

**Expanded View** for this article is available online.

## Acknowledgements

We are thankful to A. K. Dhawale for input on the experiments, data analysis, and critical reading of the manuscript. We thank A. Thrasher and D. Cox for WASP constructs, and J. Burkhardt and Nathan Roy for LifeAct-GFP mice. We thank the Advanced Imaging Center at Janelia Campus and J. Haddleston for help with the lattice light-sheet microscopy (The Advanced Imaging Center is jointly funded by the Howard Hughes Medical Institute and the Gordon and Betty Moore Foundation), and the Keck microscopy facility at the Whitehead Institute. This work was supported in part by the Koch Institute Support (core) Grant P30-CA14051 from the National Cancer Institute. We thank the Koch Institute Swanson Biotechnology Center for technical support, specifically Eliza Vasile for OMX Deltavision microscope. This work was supported by the Ragon Institute of MGH, MIT, and Harvard, the National Institutes of Health grant AI43542 (MLD) and Wellcome Trust grant PRF 100262 (MLD), support from Yale University to MM and NIH grant 5R01AI100315 to RG. DJI is an investigator of the Howard Hughes Medical Institute.

## Author contributions

SK conceived the project, performed experiments, and wrote the manuscript. MMa performed computational simulations; Y-CP, NW, and MT performed experiments; MMe, EJ, and MD provided experimental inputs; and RG edited the manuscript and provided experimental inputs and patient samples. DJI supervised the project and wrote the manuscript.

## Conflict of interest
The authors declare that they have no conflict of interest.

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
