## [Review Process File · The EMBO Journal]

Cytoskeletal tension actively sustains the migratory T cell synaptic contact

Sudha Kumari, Michael Mak, Yehchuin-Poh, Mira Tohme, Nicki Watson, Mariane Melo, Erin Janssen, Michael Dustin, Raif Geha and Darrell J. Irvine

Review timeline:

Submission date:	25th Jun 2019
Editorial Decision:	29th Jul 2019
Revision received:	30th Sep 2019
Editorial Decision:	31st Oct 2019
Revision received:	12th Nov 2019
Accepted:	18th Nov 2019

Editor: Karin Dumstrei

Transaction Report:

1st Editorial Decision

29th Jul 2019

Thank you for submitting your manuscript to The EMBO Journal. Your study has now been seen by three referees and their comments are provided below.

As you can see from the comments below, the referees find the analysis interesting and suitable for publication here. However, referees #1 and 3 also raise a number of different issues that should be resolved. I see that the points raised by referee #3 are a bit more complex to sort out, but any analysis along those lines would strengthen the findings reported.

REFeree REPORTS:

Referee #1:

Kumari et al. show that WASP dependent F-actin foci, polymerized by naïve T cells, correlate with arrested, symmetric immunological synapse maintenance. Upon WASP downregulation, T cells polarize, break synapse symmetry and resume motility. Unlike synapse formation, maintenance appears to be independent of integrins or calcium signalling. Using a combination of live imaging, IFs, traction force microscopy and modelling the authors provide evidence that the T cell actomyosin network, stabilized by WASP nucleated F-actin foci generates cytoskeletal tension at the synapse that maintains the arrested, symmetric state.

Overall the manuscript is well written and provides enough evidence to support its main hypotheses. However, the following points should be addressed.

1.) Figure 3B, traction force microscopy. Are the forces that T cells apply to the substrate balanced over time? In other words: when force is applied in one quadrant of the synapse at t0, is there a corresponding counter force at t1 in another quadrant? Otherwise I imagine the synapse would rapidly become asymmetric. Maybe it would be possible to do some cross-correlation analysis?

2.) It is great that the relevance of the findings has also been confirmed with dendritic cells. However, while it is convincing that there is certain time-window of the synapse that is integrin independent, given the conflicting evidence in the literature regarding DC-T cell interaction times (minutes to hours depending on the experimental system) it would be interesting to put this into context. Therefore it would be (a) interesting to know how long T cell-DC contacts are in the given setup and (b) if potential later phases of the interaction could become again integrin dependent or if WASP down-regulation could be attenuated in longer lasting contacts to maintain stability.

3.) The authors never really mention how they imagine force coupling to be achieved. I assume the idea is that anti-CD3 bound TCR couples to WASP foci F-actin via CD3 ζ ? As glass doesn't allow protrusion formation (mentioned in the manuscript) intercalation of protrusions is not an option.

Referee #2:

Paper from Sudha Kumari et al. describes the mechanisms involved in the stability of a symmetric synapse and how this symmetry is broken.

The authors studied T cell actin organization at different stages of activation using a combination of super-resolution imaging, genetic and pharmacological perturbations, mechanical measurements, and computational simulations. They provide important clues to how T cells maintain and break the formation of a symmetric synapse.

They describe that specialized actin microstructures termed actin foci, which form within the interface between the T cell and the APC (or APS), generate and sustain tension on the T cell actin cytoskeleton at the contact interface controlling the symmetry of the immune synapse. They show that WASP is a key player in the formation of these actin foci and that myosin IIA regulates the stability of the tension at the synapse.

Although WASP has already been involved in formation of a stable symmetrical synapse (Sims et al.) and in the regulation of forces exerted by the actin cytoskeleton, the strength of this study is to propose a computational model of the actin cytoskeletal architecture and synaptic interface mechanics. This model explain how the distribution of actin foci, and the actomyosin cytoskeleton create force tensions at the interface, which allow the formation of a symmetrical synapse. Using this model, the authors predict how decrease in the formation of actin foci or local inhibition of the myosin IIA can alter this symmetry and they test these predictions in cells.

This thorough analysis proposes a model that can be tested by the community. It will be useful to find out if all T cell populations follow the same "rules", if different APC have different ability to induce the tensile forces and how. It will also be useful to test key actin nucleators involved in actin foci formation and to describe the signaling pathways that can regulate the tension forces at the interface.

MINOR CONCERNS

Authors have mainly used anti-CD3 antibodies (and ICAM1) to induce the formation of synapses. They only used OTII CD4 T cells from wt or WASPKO mice in figure S10 and show representative images that confirm results obtained with anti-CD3 antibodies. Yet, results are not quantified.

Quantification of images with: AR, actin foci and pCasL would strengthen the data obtained with activating antibodies.

It would help the reader to have a bit more details in the figure legends (cell type, i.e. CD4+ T cells or activated OTII, some details on images shown, i.e. phalloidin or Lifeact for example)

Referee #3:

In this study, the authors examine the role of tension in symmetry breaking after the formation of immune synapses. Using advanced light microscopy and computational modeling, the authors conclude that myosin II, acting through WASP- and Arp2/3-dependent actin "foci" distributed in the innermost part of the contact between the T cell and the antigen-presenting cell, promotes en face tension that maintains the contact, even in the absence of integrin activation or calcium. The study is

interesting and reflects well the current state of the art in the field. However, several crucial points need to be addressed.

MAJOR COMMENTS:

- The nature of the actin foci is unclear. Looking at the images, they do not resemble podosomes, remaining poorly defined. A key issue is what is the signal driving these foci. The ability of the defined actin foci to recruit different proteins has not been established in comparable time points, which probably determines the functionality of each one. The molecular composition of the studied foci are based on previous reports showing WASP, pCasL and F-actin and also TCR-signaling components (Kumari et al., 2015 eLife; Janssen et al., 2016 JCI). An important issue would be then to position them with respect to TCR/CD3 components, eg. phosphorylated CD3z, PLCgamma, etc. Myosin II is observed surrounding the actin foci in the image showed, rather than forming part of the foci (Fig S12). One cannot know how authors have decided that the components used in their model are occupying in cells the places depicted (e.g.: little blue or yellow dots on their model).

- Conceptually, a major issue is symmetry breaking in these conditions. Assuming a stochastic distribution of anti-CD3 and ICAM1 in the activating surfaces, what would compel these cells to break the synapse so soon after contact? (20 min are more in the range of CTL contacts with target cells). Would other cell models of immune synapses, e.g. human CD4 T cells or Jurkat T cells behave the same?

- Along the same lines, can the authors identify external or internal factors that drive synapse breaking? This is extremely interesting and pertinent as to identify mechanisms by which synaptic contacts may be properly terminated. In the past, some of the authors have proposed that exhaustion of TCR signaling may drive this process. Is this the case here?

- A major issue with this study is that it assumes that antigen-presenting surfaces are infinitely hard (they address this using hydrogels, so this is not an issue) and infinitely planar (which is not really addressed). In the real context in lymph nodes, APCs are mainly dendritic cells, which are slightly larger than T cells, but are not planar at all, presenting curved surfaces, grooves, creases, etc. In addition, DCs are more or less as pliable as T cells. In other words, it is possible that the reported phenomena are extreme cases in which the APS is much larger than the interactive surface of the T cell? Although this is a very hard issue to tackle, it is paramount to understand if these foci/ lattices are physiologically relevant or not. The authors need to address this issue experimentally and in their simulations.

OTHER SPECIFIC COMMENTS:

- The meaning of the AR parameter is not correctly stated in the text. One has to guess whether its value is to be higher than 1 or not for asymmetry/symmetry (provided this value has a specific meaning).

- The azidoblebbistatin inhibitor shows identical myosin inhibition properties in absence of U.V. irradiation, in vitro and in vivo, than blebbistatin does. Upon sequential pulses of 350 nm irradiation (and probably 450 nm, as used here) it is cross-linked to the protein, leading to a stable inhibited state of myosin; also, free azidoblebbistatin seems to be inactivated by these pulses (Kepiro et al, PNAS 2012). This means that myosin II is inhibited in the presence of azidoblebbistatin in a reversible manner until U.V. covalently links to it upon irradiation or in areas where the inhibitor has not been irradiated. How long was the azidoblebbistatin added to the cells until UV irradiation? The azidoblebbistatin present on the plate but not irradiated is supposedly acting also on the "control area". Indeed, it is not clear why authors claim that the actin cytoskeleton is cross-linked by the action of the inhibitor (page 10). The inhibitor is cross-linked to the myosin motor and inactivates it in an irreversible manner; this does not mean that it is cross-linked to the actin filaments. It would have been preferably that authors wash the rest of the inhibitor in their experiments, after the U.V. irradiation, to avoid this part of the story where the inhibitor acts as blebbistatin. Also, additional evidence of actomyosin crosslinking in the photostimulated regions should be presented.

- Regarding the same issue (Fig. 5H), the UV is directed to the external actin lamella; where there are no or virtually no actin foci. Is this correct? The correct control conditions here must include

vehicle and blebbistatin, if azidoblebbistatin is not washed from the medium.

- In Fig 2B the graph shows normalized values for AR and Foci. How was normalization performed and which was the meaning of each point? It is quite difficult to assess results text and data correspondence in the figures (general comment for the whole manuscript).

- In Fig 2A and 2F-G, the F-actin foci do not move at all (as shown in the kymograph). This result does not correspond to those shown in Kumari et al 2015, in which a fraction of the F-actin foci moved centripetally together with TCR clusters. Therefore, "dynamics" does not correspond precisely with the meaning intended here, and the term "kinetics" would be more accurate.

- In Fig 6 the results shown with pCasL, actin and WASP were already known. Fig 6C shows that in presence of softer substrates WT cells are more symmetric and that when increasing stiffness WT cells are more similar to WASP deficient cells, although still very different? WASP deficient cells show a decrease in the AR?? Are they more symmetric when stiffness increases? Stiffness in text (2.5 kPa and 100 kPa; page 11) does not correspond with figure (2.5 and 50 kPa).

- Is pCasL related to Myosin and how? How do the authors link all their results with previous and current data? How can WASP deficient cells achieve CD3 central localization (see Fig 6A)? What are the clusters of F-actin inside the cell in Fig 6D, WASP-/-?

Minor comment:

- Complete the reference: Bohineust et al., 2018

1st Revision - authors' response

30th Sep 2019

Please see next page.

Referee #1:

Kumari et al. show that WASP dependent F-actin foci, polymerized by naïve T cells, correlate with arrested, symmetric immunological synapse maintenance. Upon WASP downregulation, T cells polarize, break synapse symmetry and resume motility. Unlike synapse formation, maintenance appears to be independent of integrins or calcium signaling. Using a combination of live imaging, IFs, traction force microscopy and modelling the authors provide evidence that the T cell actomyosin network, stabilized by WASP nucleated F-actin foci generates cytoskeletal tension at the synapse that maintains the arrested, symmetric state.

Overall the manuscript is well written and provides enough evidence to support its main hypotheses.

We thank the reviewer for these comments.

However, the following points should be addressed.

1.) Figure 3B, traction force microscopy. Are the forces that T cells apply to the substrate balanced over time? In other words: when force is applied in one quadrant of the synapse at t_0 , is there a corresponding counter force at t_1 in another quadrant? Otherwise I imagine the synapse would rapidly become asymmetric. Maybe it would be possible to do some cross-correlation analysis?

The reviewer raises an interesting point. First, we should point out that traction force microscopy (TFM) analysis operates on the assumption that forces are balanced, and the net force is zero (Maruthamuthu V et al, Proc Natl Acad Sci U S A. 2011; Schwarz US, Biochem Biophys Acta. 2015). As the reviewer correctly points out, if this were not the case, the cell would accelerate. However, an interesting question raised by the reviewer is whether forces are balanced via a dipole like distribution at opposing quadrants in the cell, or are contact forces evenly spread out across the synapse? We attempted answering this question by calculating the net contractile moment (Butler et al., Am J Physiol Cell Physiol 282: C595–C605, 2002) in the synapse. The net contractile moment is a scalar measure of the cell's contractile strength, and represents a force generator comprising only two imaginary point forces in the cell that are equal, opposite, and separated by a distance (Wang N et al., Am J Physiol Cell Physiol. 2002 Mar;282(3):C606-16). The larger the value of the net contractile moment, the more dipole-like is the distribution of traction force in the cell. We analyzed the contractile force moment in the synapse by averaging the traction forces around two highest traction zones in the interface, and multiplying them by the distance separating them. The results reveal much lower contractile moments in both WT and WASP^{-/-} cells compared to that described previously in fibroblasts, suggesting that the traction forces are more evenly distributed than dipole-like in T cells. The higher moment in WT synapses compared to WASP^{-/-} synapses

(shown below), indicates that these cells are likely to have stronger and more distantly separated opposing forces generated in the interface than the WASP^{-/-} cells, just as the reviewer predicted. We have now included this new data in the manuscript (**page 6, paragraph 3, graphed below for the reviewers**). While further analysis of force distributions in naïve T cell synapse is interesting and pertinent, this is technically challenging, as it would require specialized image and traction analysis methodologies to accommodate the small size (diameter ~5µm) and fast temporal dynamics of these cells, which are beyond the scope of the present paper.

Reviewer's Figure. 1. Higher contractile moment in WT synapse signifies higher opposing point forces separated by longer distances.

It is great that the relevance of the findings has also been confirmed with dendritic cells. However, while it is convincing that there is certain time-window of the synapse that is integrin independent, given the conflicting evidence in the literature regarding DC-T cell interaction times (minutes to hours depending on the experimental system) it would be interesting to put this into context. Therefore it would be (a) interesting to know how long T cell-DC contacts are in the given setup and (b) if potential later phases of the interaction could become again integrin dependent or if WASP down-regulation could be attenuated in longer lasting contacts to maintain stability.

Indeed, there is conflicting evidence on the interaction times (ranging from minutes to hours) of CD4⁺ T cells with DCs (Donnadieu et al., Current Biology, 1994; Gunzer et al., Immunity, 2000; Benvenuti et al., J.Immunology, 2004; Celli et al., Immunity, 2007). To characterize the duration of CD4⁺ T cell-DC synapses in our system, we have performed time course experiments of naïve CD4⁺ T cells with bone marrow derived dendritic cells. During these experiments, we monitored levels of activated (phospho-tyr 293) WASP, actin foci as well as the aspect ratio of T cells activated on DCs. These experiments revealed that similar to experiments carried out using anti-CD3/ICAM-coated coverslips (antigen-presenting surfaces, APS), the duration of DC-T cell synapses in our system was on the order of 20 minutes, with downregulation of phospho-WASP and actin foci between 10 and 20 minutes, and statistically significant increases in T cell aspect ratios by 20 minutes. We have included this new data in the revised version of the manuscript (**new Figure**

S5). In addition, we found that WASP overexpression prolongs synapse symmetry even at 20 min after initial T cell-APC interactions, indicating that WASP levels are crucial for synapse symmetry at later phases of the synapse as well. This new data is also included in the revised version of the manuscript (**new Figure S7A, B**). We have commented on these new findings in the results (**page 6, paragraph 2; page 7, paragraph 2**).

Examining integrin function at the T cell-BMDC synapse is challenging, since we would need to inhibit LFA-1 in a T cell selective fashion and the major T cell integrin responsible for T cell synapse formation, LFA-1, is also expressed by DCs (Balkow, Blood 2010). Furthermore, a recent report showed that CD4⁺ T cells do not require ICAM-1 (ligand for LFA-1) to form functional synapses with DCs in lymph nodes (Feigelson et al., Cell Reports, 2018). Therefore, the stability of CD4⁺ T cell synapses *in vivo* is also likely to be integrin independent. We have included this point in the discussion section (**page 12, paragraph 3**).

3.) The authors never really mention how they imagine force coupling to be achieved. I assume the idea is that anti-CD3 bound TCR couples to WASP foci F-actin via CD3ζ? As glass doesn't allow protrusion formation (mentioned in the manuscript) intercalation of protrusions is not an option.

*We apologize for not explaining our model of force coupling better. As the reviewer points out, synapse stability does not depend on protrusions. In our model, the in-plane actomyosin forces responsible for synapse stability assert pulling forces on the substrate via membrane receptors. Although we have not confirmed the identity of these receptors, they are likely to be TCR microclusters. We have clarified this proposed mechanism in the revised version of the manuscript (**Figure 6 legend 'Model', page 24, paragraph 4**).*

Referee #2:

Paper from Sudha Kumari et al. describes the mechanisms involved in the stability of a symmetric synapse and how this symmetry is broken. The authors studied T cell actin organization at different stages of activation using a combination of super-resolution imaging, genetic and pharmacological perturbations, mechanical measurements, and computational simulations. They provide important clues to how T cells maintain and break the formation of a symmetric synapse. They describe that specialized actin microstructures termed actin foci, which form within the interface between the T cell and the APC (or APS), generate and sustain tension on the T cell actin cytoskeleton at the contact interface controlling the symmetry of the immune synapse. They show that WASP is a key player in

the formation of these actin foci and that myosin IIA regulates the stability of the tension at the synapse.

Although WASP has already been involved in formation of a stable symmetrical synapse (Sims et al.) and in the regulation of forces exerted by the actin cytoskeleton, the strength of this study is to propose a computational model of the actin cytoskeletal architecture and synaptic interface mechanics. This model explain how the distribution of actin foci, and the actomyosin cytoskeleton create force tensions at the interface, which allow the formation of a symmetrical synapse. Using this model, the authors predict how decrease in the formation of actin foci or local inhibition of the myosinIIa can alter this symmetry and they test these predictions in cells. This thorough analysis proposes a model that can be tested by the community. It will be useful to find out if all T cell populations follow the same "rules", if different APC have different ability to induce the tensile forces and how. It will also be useful to test key actin nucleators involved in actin foci formation and to describe the signaling pathways that can regulate the tension forces at the interface.

We thank the reviewer for the encouraging assessment. In this paper we have largely looked at the mechanical function of actin foci in synapse stability, and have not followed the how these principles alter in different T cell subtypes (Colin-York et al, JCS, 2019), or, how these principles could consolidate the impact APC stiffness has on T cell activation (Blumenthal et al., BioArxiv, 2019; <https://doi.org/10.1101/680132>). Indeed, a These are all excellent points and we hope to investigate at least some of them in the future.

MINOR CONCERNS

Authors have mainly used anti-CD3 antibodies (and ICAM1) to induce the formation of synapses. They only used OTII CD4 T cells from WT or WASPKO mice in Figure S10 and show representative images that confirm results obtained with anti-CD3 antibodies. Yet, results are not quantified. Quantification of images with: AR, actin foci and pCasL would strengthen the data obtained with activating antibodies.

We thank the reviewer for pointing this out. We have provided quantification of AR, pCasL and actin foci in the **Figure S10A** in the revised version of the manuscript. In addition, we have carried out additional experiments characterizing WASP and actin foci dynamics in CD4+ T cells interacting with bone marrow-derived dendritic cells (**new Figure S5**), which support the conclusions obtained with anti-CD3/ICAM-1-coated surfaces.

It would help the reader to have a bit more details in the Figure legends (cell type, i.e. CD4+ T cells or activated OTII, some details on images shown, i.e. phalloidin or Lifeact for example)

We have now provided the aforementioned details in the revised version of the manuscript.

Referee #3:

In this study, the authors examine the role of tension in symmetry breaking after the formation of immune synapses. Using advanced light microscopy and computational modeling, the authors conclude that myosin II, acting through WASP- and Arp2/3-dependent actin "foci" distributed in the innermost part of the contact between the T cell and the antigen-presenting cell, promotes en face tension that maintains the contact, even in the absence of integrin activation or calcium. The study is interesting and reflects well the current state of the art in the field. However, several crucial points need to be addressed.

MAJOR COMMENTS:

- The nature of the actin foci is unclear. Looking at the images, they do not resemble podosomes, remaining poorly defined. A key issue is what is the signal driving these foci. The ability of the defined actin foci to recruit different proteins has not been established in comparable time points, which probably determines the functionality of each one. The molecular composition of the studied foci are based on previous reports showing WASP, pCasL and F-actin and also TCR-signaling components (Kumari et al., 2015 eLife; Janssen et al., 2016 JCI). An important issue would be then to position them with respect to TCR/CD3 components, eg. phosphorylated CD3z, PLCgamma, etc.

As the reviewer correctly points out, actin foci have been previously characterized at much earlier stages of the synapse than the focus of our study (Kumari et. al., eLife 2015; Hashimoto-Tane et. al., J.Exp. Med., 2016). However, at the later stages of synapse the foci are still WASP-dependent: In new experiments, we forced WASP overexpression at late times and show that this rescues actin foci and blocks cell polarization at 20 minutes (**new Figure S7A**). To further address the reviewer's questions, have measured the colocalization of foci with TCR microcluster components: phospho-Zap-70 (Tyr319)/Syk (Tyr352) and PLCy1 Phospho-PLCy1 (Tyr783) to ascertain whether foci play a signaling role in the late synapse, similar to that described in early synapses (Kumari et al., eLife, 2015). We find that while these components are highly colocalized with actin foci in early (5') synapses, in the late synapses the foci have significantly lower colocalization with both phospho-Zap70 and phospho-PLCy1. These data indicate that the signaling and mechanical roles of foci are not conserved throughout the lifetime of the synapse, and actin foci may not have a signaling function in the late synapse. We have included this data in the revised version of the manuscript (**new Figure S9B, C; page 7, paragraph 3**).

Myosin II is observed surrounding the actin foci in the image showed, rather than forming part of the foci (Fig S12). One cannot know how authors have decided that the components used in their model are occupying in cells the places depicted (e.g.: little blue or yellow dots on their model).

We did not predetermine the spatial localization of any component in our model except for the sites where foci would eventually form. Instead the spatial patterns noticed by the reviewer emerged as a result of the Brownian dynamics and inter-component interactions in the simulation. Initially myosin molecules were randomly distributed in the simulation space. As the simulations progress, myosin was enriched at the foci as well as the inter-foci regions. The enrichment of myosin at actin foci in the simulation could be due to higher actin concentration at the foci than the surrounding lamellae, as established previously (Kumari et al., eLife, 2015). At this stage, we do not fully understand why we do not observe localization of myosin at foci in our experiments. The complex higher order spatial patterning effect of Myosin II may underlie this effect but uncovering its mechanistic underpinnings is currently beyond the scope of our study. One possibility is that due to constant filament nucleation, foci-recruited myosin molecules tend to move away from foci, and the limited number that remain may be below the detection limit of fluorescence microscopy. We have commented on these issues in the **caption of new Figure S14 A**.

- Conceptually, a major issue is symmetry breaking in these conditions. Assuming a stochastic distribution of anti-CD3 and ICAM1 in the activating surfaces, what would compel these cells to break the synapse so soon after contact? (20 min are more in the range of CTL contacts with target cells). Would other cell models of immune synapses, e.g. human CD4 T cells or Jurkat T cells behave the same?

The synapse duration described here is well within the range of the synapse lifetimes measured previously for naïve CD4⁺ T cells interacting with dendritic cells *in vitro* (Donnadieu, Current Biology, 2004; Gunzer, Immunity, 2000; Benvenuti, J. Immunology, 2004). However, the reviewer raises an interesting question about the mechanisms underlying symmetry breaking. Symmetry breaking and motility progression using lamellar protrusions is a fundamental feature of T cell behavior since it is required for effective immunosurveillance. Our study suggests that maintaining a stable synapse for prolonged period requires the active formation of actin foci by WASP to stabilize the synapse against lamellar fluctuations. These fluctuations are a constant and ongoing phenomenon in the T cell cortex (Roybal et al., Immunological Reviews, 2013; Sims et al., Cell, 2007; Ritter et al., Immunity, 2015; Krummel et al., Curr. Opin. Cell Biol., 2014), and are also seen during the synaptic phase (Figure 2E). Consequently, the synapse would destabilize if WASP was degraded and new actin foci ceased to

form. The clock for WASP/ foci degradation and subsequent symmetry breaking is likely set by the cellular machinery that degrades WASP, perhaps in sync with the degradation of TCR signaling components. In line with this hypothesis, previous studies have described a global protein degradation program that is triggered by T cell receptor activation (Wiedemann A. et. al., Immunology Letters, 2005; Huang and Gu, Immunology Letters, 2008; O’Leary et. al., Front. Immunol. 2015; Friend et. al, Am J Clin Exp Immunol, 2014; Heissmeyer et. al., Nature Immunology, 2004). WASP is a substrate of this program (Macpherson et al., Hematologica, 2012; Reicher et al., MCB, 2012; Watanabe et al., AJACI, 2013). We have elaborated on these points in the ‘Discussion’ section (**page 12, paragraph 4**).

Human naive CD4⁺ T cells are also likely to disassemble synapses using similar mechanisms as their murine counterparts since WASP is also required for synapse sustenance in human CD4⁺ T cells (Figure 3D; Figure S10B). To further confirm this, we monitored synapse symmetry breaking in primary human CD4⁺ T cells, and find that the basic tenets (low phospho-WASP, pCasL, foci), as well as the lifetime of synapse is comparable between human and mouse primary CD4⁺ T cells. We have included this new data in the revised version of the manuscript (**new Figure S7C**).

- Along the same lines, can the authors identify external or internal factors that drive synapse breaking? This is extremely interesting and pertinent as to identify mechanisms by which synaptic contacts may be properly terminated. In the past, some of the authors have proposed that exhaustion of TCR signaling may drive this process. Is this the case here?

Indeed, it may very well be the case that downregulation of TCR signaling leads to WASP degradation and synapse disassembly, since WASP degradation (potentially via TCR-dependent mechanisms) accompanies synapse symmetry breaking (Figure 2C-E). Our model hypothesizes that the migration is the “default” mode in the naïve T cells, and WASP-based foci program is required to restrain the motility. Thus, the trigger for motility is cell-intrinsic migratory potential. In support of this hypothesis, overexpression of WASP using a lentiviral system prevents symmetry breaking in late synapses. We have included these new results in the revised version of the manuscript (**new Figure S7A, B**).

- A major issue with this study is that it assumes that antigen-presenting surfaces are infinitely hard (they address this using hydrogels, so this is not an issue) and infinitely planar (which is not really addressed). In the real context in lymph nodes, APCs are mainly dendritic cells, which are slightly larger than T cells, but are not planar at all, presenting curved surfaces, grooves, creases, etc. In addition, DCs are more or less as pliable as T cells. In other words, it is possible that the reported phenomena are extreme cases in which the APS is much larger than the interactive surface of the T cell? Although

this is a very hard issue to tackle, it is paramount to understand if these foci/ lattices are physiologically relevant or not. The authors need to address this issue experimentally and in their simulations.

Although we only ran our simulations on planar surfaces, this is by no means a fundamental assumption of our model since the simulation parameters can be extended to a curved plane. This is because the F-actin lattice does not have to be planar but can instead be curved along a manifold, and would still form a tensile system that prevents radial symmetry breaking. To confirm this, we have examined F-actin patterning on a curved surface in our simulations, and find it to be in agreement with the simulations on the flat surface (**new Figure S14B; page 9, paragraph 2**). We used 9.6 μ m as the radius of the sphere that our surface is curved onto, which is on the scale of DCs. When assembled on a curved surface, the actin network still formed a lattice with actin concentrated at foci and interconnections between foci across the interface (**Figure S14B**). The subtle, higher order effects of varying curvatures, however, are beyond the scope of this current study, as a detailed systematic analysis with novel metrics that can account for curved surfaces need to be developed.

In addition, we carried out experiments to evaluate if WASP downregulation correlated with loss of actin foci and cell morphological polarization in live T cell-dendritic cell contacts. As shown in new Figure S5, murine CD4⁺ T cells interacting with bone marrow derived dendritic cells show onset of polarization coincident with WASP and actin foci downregulation (**new Figure S5**). Thus, we believe the findings in the model system accurately capture the key features of bona fide T cell-APC synapses.

OTHER SPECIFIC COMMENTS:

- The meaning of the AR parameter is not correctly stated in the text. One has to guess whether its value is to be higher than 1 or not for asymmetry/symmetry (provided this value has a specific meaning).

We thank the reviewer for pointing this out. We have now fixed this in the text (**page 4, paragraph 2; Figure 1A**).

- The azidoblebbistatin inhibitor shows identical myosin inhibition properties in absence of U.V. irradiation, in vitro and in vivo, than blebbistatin does. Upon sequential pulses of 350 nm irradiation (and probably 450 nm, as used here) it is cross-linked to the protein, leading to a stable inhibited state of myosin; also, free azidoblebbistatin seems to be inactivated by these pulses (Kepiro et al, PNAS 2012). This means that myosin II is inhibited in the presence of azidoblebbistatin in a reversible manner until U.V. covalently links to it upon irradiation or in areas where the inhibitor has not been irradiated.

How long was the azidoblebbistatin added to the cells until UV irradiation? The azidoblebbistatin present on the plate but not irradiated is supposedly acting also on the "control area". Indeed, it is not clear why authors claim that the actin cytoskeleton is cross-linked by the action of the inhibitor (page 10). The inhibitor is cross-linked to the myosin motor and inactivates it in an irreversible manner; this does not mean that it is cross-linked to the actin filaments. It would have been preferably that authors wash the rest of the inhibitor in their experiments, after the U.V. irradiation, to avoid this part of the story where the inhibitor acts as blebbistatin. Also, additional evidence of actomyosin crosslinking in the photostimulated regions should be presented.

We apologize for the confusion. Indeed, azidoblebbistatin crosslinks with myosin II motors, and not the actomyosin cytoskeleton, following UV irradiation. We will amend the text to rectify this. This change does not affect our interpretations, since our simulations only assumed that only that myosin activity was inhibited by azido-blebbistatin photocrosslinking and did not assume the actin network to be cross-linked. In fact, cross-linking of the network by azido-blebbistatin in the photoactivated area would not have led to tension decay and symmetry breaking in the simulation.

The reviewer also hypothesizes that the azido-blebbistatin outside of the photocrosslinked area would behave like blebbistatin and would therefore inhibit myosin in "control" regions outside the cross-linked area. We believe that this should not be the case for two reasons. First, we used a low concentration of azido-blebbistatin (5 μ M) which is 10 times lower than the concentration of Blebbistatin (50 μ M) routinely utilized in the literature for myosin inhibition (Cell free IC_{50} = 2.5 μ M; Kepiro et al., *Angewandte Chemie*, 2012; Babich et al., *JCB*, 2012). Thus azido-blebbistatin is unlikely to inhibit myosin unless the two are cross-linked to each other. Second, we utilized 488 nm laser to image the cell during the course of symmetry breaking. 488 nm excitation light is known to inactivate free azido-blebbistatin, thereby rendering the effective concentration of azidoblebbistatin even less than 5 μ M in the "control area". At such low concentrations, azido-blebbistatin is unlikely to be effective.

It is not possible, as the reviewer suggests, to wash out the inhibitor during imaging since the time window between photocrosslinking and deflection of the center of mass of the cell is far too short (within 5 sec, Figure 5I). Instead, we have provided data showing that without UV crosslinking, 5 μ M azido-blebbistatin does not cause symmetry breaking in our T cell imaging conditions. We have included this new data in the revised version of the manuscript (**new Figure S15**).

- Regarding the same issue (Figure 5H), the UV is directed to the external actin lamella; where there are no or virtually no actin foci. Is this correct? The correct control conditions here must include vehicle and blebbistatin, if azidoblebbistatin is not washed from the medium.

The foci continue to form and dissipate throughout the synaptic interface including the synapse periphery (Videos S3, S4, S10). Thus, the foci would form in the peripheral photo cross-linked zone. The foci may be difficult to ascertain in the images because we utilized very low excitation power for the laser (0.5 mW) for imaging actin in these experiments so as to minimize photodamage to the cell. We have now provided the data showing that 5 μ M azido-blebbistatin by itself does not cause symmetry breaking, as mentioned above (**new Figure S15**).

- In Figure 2B the graph shows normalized values for AR and Foci. How was normalization performed and which was the meaning of each point? It is quite difficult to assess results text and data correspondence in the figures (general comment for the whole manuscript).

In Figure 2B each point represents a time point during live imaging. The data points for AR and foci were normalized to their mean values and plotted to enable display on the same graph. We have amended the text and figure legends to clarify the normalization schemes used.

- In Figures 2A and 2F-G, the F-actin foci do not move at all (as shown in the kymograph). This result does not correspond to those shown in Kumari et al 2015, in which a fraction of the F-actin foci moved centripetally together with TCR clusters. Therefore, "dynamics" does not correspond precisely with the meaning intended here, and the term "kinetics" would be more accurate.

The reason why the foci moved in Kumari et. al. was because the aforementioned study utilized lipid bilayers to activate T cells, where TCR microclusters (and associated foci) are mobile. In contrast, T cells in the present study were activated on substrates or APCs that do not allow for lateral movement of TCR microclusters within the synapse (Brossard et al, Eur. J. Immunology, 2005). As such, motility of TCR and associated foci does not impact foci-dependent synapse symmetry, as long as the foci have longer lifetime than the lamellar fluctuations (Figure 6A, **now elaborated on in discussion, page 13, paragraph 2**).

We have replaced the term "dynamics" with "kinetics" in the relevant section of the text.

- In Figure 6 the results shown with pCasL, actin and WASP were already known. Figure 6C shows that in presence of softer substrates WT cells are more symmetric and that when increasing stiffness WT cells are more similar to WASP deficient cells, although still very different? WASP deficient cells show

a decrease in the AR?? Are they more symmetric when stiffness increases? Stiffness in text (2.5 kPa and 100 kPa; page 11) does not correspond with figure (2.5 and 50 kPa).

We agree with the reviewer that there seems to be less of a shape difference between the WT and WASP^{-/-} cells on stiffer substrates, although the difference in AR is statistically significant to a similar extent in both cases. We do not understand the reason for this phenomenon at the moment. One possibility is higher integrin activation on harder substrates (Comrie et al., J.C.B., 2015), could facilitate synapse breaking and shape elongation in WT T cells increasing their AR marginally.

We thank the reviewer for pointing out the error in the text. The hydrogels are 100 KPa, as mentioned in the figure. We have amended the text to reflect that.

Realizing that we have shown the WASP^{-/-} T cells-APC data already in the manuscript (Figure S12, S13), we have now removed the (D) panel from Figure 6 to minimize redundancy. In the previous version of the manuscript, this panel showed higher AR of WASP^{-/-} cells activated on APC.

- Is pCasL related to Myosin and how? How do the authors link all their results with previous and current data? How can WASP deficient cells achieve CD3 central localization (see Fig 6A)? What are the clusters of F-actin inside the cell in Fig 6D, WASP^{-/-}?

The relationship between pCasL, actomyosin cytoskeleton and tension in fibroblasts and T cells has been well established elsewhere (Sawada et. al., Cell, 2006; Santos et. al., ICB, 2016, Janssen et. al., JCI, 2016). CasL does not directly interact with myosin; rather CasL phosphorylation is exquisitely sensitive to the tension that myosin activity generates in the actin network. We have included a molecular schematic to show CasL phosphorylation in the context of immunological synapse (Figure S5 A, B). We have elaborated on this point in the revised version of the manuscript (**caption Figure S6**).

Indeed WASP-deficient cells can achieve TCR microcluster colocalization in the central zone (Kumari et. al. eLife). It is likely to be via a WASP-independent pool of F-actin such as the filaments nucleated via formins (Murugesan et al., JCB, 2016), or via microtubule motor dyneins (Hashimoto-Tane, Immunity, 2011), both of which have been shown to be a mode of transport for TCR microclusters at the synapse. We have included these explanations in the discussion section of the revised version of the manuscript (**page 13, paragraph 2**).

The dim clusters of F-actin within the cell shown in Figure 6D, we believe, could be the cluster of microvilli-like surface features that T cells display in profusion prior to activation (Kim et al., Nature Communications, 2018).

Minor comment:

- Complete the reference: Bohineust et al., 2018

This has been corrected.

2nd Editorial Decision

31st Oct 2019

Thank you for sending us your revised manuscript. Your manuscript has now been re-reviewed by referee #1 and the comments provided below. As you can see the referee appreciates the introduced changes and support publication here. I am therefore very pleased to let you know that we are happy to accept your manuscript.

Before I can send you the formal acceptance letter the following editorial points need to be taken care of.

REFeree REPORTS:

Referee #1:

The authors responded adequately to my questions and improved the manuscript so that I consider it ready for publication.

2nd Revision - authors' response

12th Nov 2019

The authors performed the requested changes.

3rd Editorial Decision

18th Nov 2019

Thanks for submitting your revised manuscript to The EMBO Journal. I have had a chance to take a careful look at everything and all looks good.

I am therefore very pleased to accept the manuscript for publication here.

Corresponding Author Name: Sudha Kumari, Darrell J. Irvine

Journal Submitted to: EMBO J.

Manuscript Number: EMBOJ-2019-102783